# The All-Seeing Project: Towards Panoptic Visual Recognition and Understanding of the Open World

**Weiyun Wang**[*,1,2], **Min Shi**[*,1,3], **Qingyun Li**[*,1,4], **Wenhai Wang**[*,1], **Zhenhang Huang**[*,1],
**Linjie Xing**[*,1], **Zhe Chen**[1,5], **Hao Li**[1,6], **Xizhou Zhu**[1,7], **Zhiguo Cao**[3], **Yushi Chen**[4],
**Tong Lu**[5], **Jifeng Dai**[†,1,8], **Yu Qiao**[1]

[1]OpenGVLab, Shanghai AI Laboratory, [2]Fudan University,
[3]Huazhong University of Science and Technology, [4]Harbin Institute of Technology,
[5]Nanjing University, [6]The Chinese University of Hong Kong
[7]SenseTime Research, [8]Tsinghua University

## Abstract

We present the All-Seeing (AS)[1] project: a large-scale dataset and model for recognizing and understanding everything in the open world. Using a scalable data engine that incorporates human feedback and efficient models in the loop, we create a new dataset (AS-1B) with over 1.2 billion regions annotated with semantic tags, question-answering pairs, and detailed captions. It covers a wide range of 3.5 million common and rare concepts in the real world and has 132.2 billion tokens that describe the concepts and their attributes. Leveraging this new dataset, we develop the All-Seeing model (ASM), a unified framework for panoptic visual recognition and understanding. The model is trained with open-ended language prompts and locations, which allows it to generalize to various vision and language tasks with remarkable zero-shot performance, including both region- and image-level retrieval, region recognition, captioning, and question-answering. We hope that this project can serve as a foundation for vision-language artificial general intelligence research. Code is available at https://github.com/OpenGVLab/all-seeing.

## 1 Introduction

Creating artificial general intelligence (AGI) systems that can match human intelligence and excel in any task across domains is the ultimate goal of artificial intelligence. Recent advancements in Large Language Models (LLMs) have demonstrated impressive zero-shot capabilities in user-tailored natural language processing (NLP) tasks, suggesting new avenues for achieving AGI. However, as shown in Fig. 1a, most popular LLMs (OpenAI, 2022; Touvron et al., 2023; Chiang et al., 2023) are limited to processing language information and lack the ability to understand the visual world. Although there have been some recent developments (Zhu et al., 2023b; Liu et al., 2023a; Li et al., 2023a; Dai et al., 2023) in open-world visual understanding, they are primarily focused on understanding images as a whole, rather than recognizing and comprehending individual instances within the scene (see Fig. 1b). *This goes against the nature of the human visual system*, as described by the feature integration theory (Treisman & Gelade, 1980), which suggests that we attentively gather visual features and contexts in certain regions to achieve high-level understanding and recognition, rather than analyzing all information simultaneously.

To achieve instance-level visual understanding like humans, there are two major challenges as follows: (1) *The scarcity of open-world instance-text pair data.* As listed in Table 1, existing datasets, such as Visual Genome (Krishna et al., 2017), have limitations in terms of data scale. Laion-5B (Schuhmann et al., 2022) only contains web-crawled image-text pairs without location information, and SA-1B (Kirillov et al., 2023) lacks semantic information. (2) *The lack of spatial information modeling in most existing models*. These models mainly focus on whole-image understanding. In this work, we

---

[*]Equal contribution. This work is done when Weiyun Wang, Min Shi, and Qingyun Li are interns at Shanghai AI Laboratory. [†]Corresponding to Jifeng Dai <daijifeng@tsinghua.edu.cn>.

[1]"All-Seeing" is derived from "The All-Seeing Eye", which means having complete knowledge, awareness, or insight into all aspects of existence.

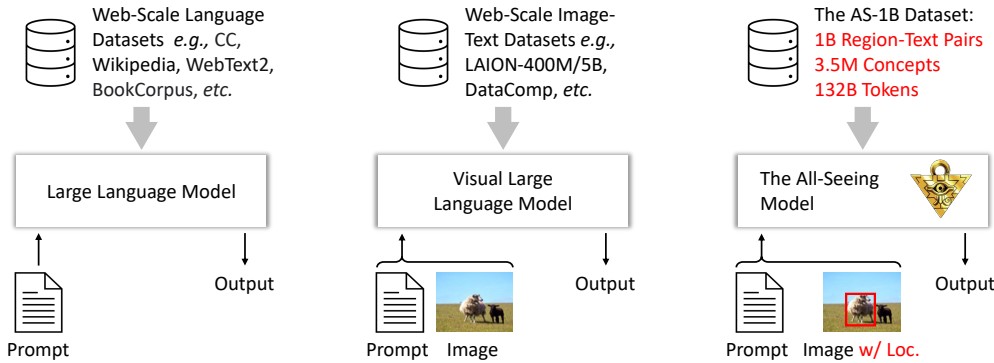

(a) LLMs possess extensive world knowledge and demonstrate impressive reasoning capabilities, but lack the ability to receive and comprehend visual information.

(b) VLLMs can process both text and images, but they can only capture the holistic visual information of the whole image and understand it based on LLMs.

(c) Our ASM can comprehensively recognize and understand the objects or concepts in regions of interest while maintaining the capabilities of VLLMs and LLMs.

Figure 1: **Overview and comparison of our All-Seeing project with other popular large foundation models.** To address the limitations of LLMs in understanding visual inputs and VLLMs in effectively leveraging region-aware information, we propose (1) a large-scale dataset AS-1B which consists of over 1 billion region-text pairs, 3.5 million open-world concepts, and 100 billion tokens of region-related question-answering and caption; and (2) the All-Seeing model (ASM), which is capable of recognizing and understanding context in arbitrary regions.

propose the All-Seeing (AS) project for open-world panoptic visual recognition and understanding, driven by the goal of creating a vision system that mimics human cognition. The term "panoptic" refers to including everything visible in one view (Kirillov et al., 2019). The AS project addresses the challenges from both the data and model perspectives.

**From the data aspect**, we propose the All-Seeing 1B (AS-1B) dataset, consisting of over 1.2 billion region annotations in various formats, such as semantic tags, question-answering pairs, and detailed captions (refer to Fig. 2). AS-1B dataset is made possible by a scalable semi-automatic data engine, which significantly lowers the previously unaffordable expense of manually annotating a massive amount of open-world semantics. The data engine operates in a "data-human-model" loop, iteratively refining data quality. Initially, diverse models, including large language models (LLMs) (Chiang et al., 2023), detection (Wang et al., 2023c; Fang et al., 2023a), captioning (Li et al., 2022a), and visual question answering (VQA) models (Liu et al., 2023a; Zhu et al., 2023b; Liu et al., 2023b), are employed as "annotators", which add semantic annotations to dense region proposals generated by state-of-the-art object detectors (Kirillov et al., 2023; Fang et al., 2023a; Li et al., 2022b; Wang et al., 2023c). Subsequently, human annotators verify the generated pseudo labels and provide feedback with high-quality data, which is then used to fine-tune the models to improve their performance. The enhanced models are then used to re-annotate the data, starting another iteration of the loop. As shown in Fig. 2, AS-1B contains a wide range of open-world concepts, including over 3.5 million different semantic tags ranging from common categories (*e.g.*, human, backpack) to fine-grained or rare categories with attributes (*e.g.*, old metal latches). AS-1B also encompasses the 3.3 billion visual question-answering pairs and 1.2 billion region captions for 1.2 billion regions.

**In terms of the model perspective**, we propose the All-Seeing model (ASM), a unified location-aware image-text foundation model. The model consists of two key components: a location-aware image tokenizer and an LLM-based decoder. The location-aware image tokenizer uses location information such as bounding box as conditions (see Fig. 1c) to extract image features, which contribute to the location capability of ASM. The LLM-based decoder inherits the world knowledge and reasoning capability from LLMs, providing a strong foundation for visual recognition and understanding. In addition, to unify image-text aligning and generation tasks, we introduce a new decoding approach, where the aligning tasks are reformulated as a "special" generation task, enabling our model to generalize to various vision-language tasks with shared weights. Compared to previous methods (Radford et al., 2021; Li et al., 2023a; Liu et al., 2023a; Zhu et al., 2023b), our work offers several advantages as follows: (1) Our model not only excels in image-level understanding but also demonstrates exceptional capability in recognizing and comprehending objects at the instance

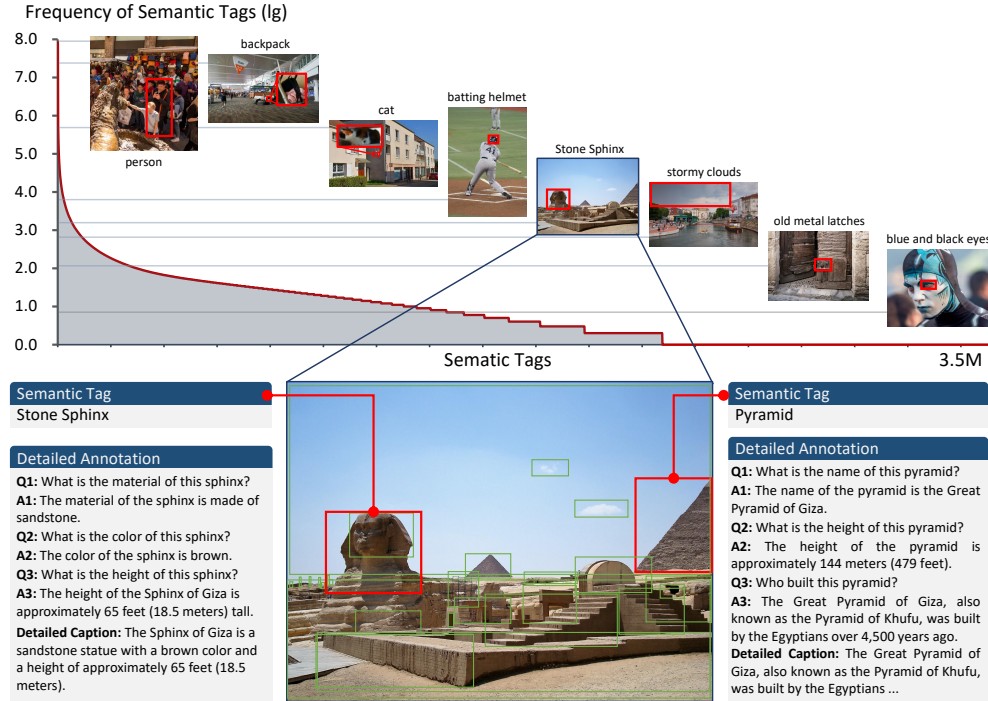

Figure 2: **Semantic concepts and annotations in the AS-1B dataset**. The semantic tags in AS-1B dataset encompass a wide range of concepts. Beyond brief semantic tags, detailed annotations, including visual-question-answering pairs and region captions are also provided.

level, closely aligning with human cognitive processes. (2) Our model is a unified framework that supports a wide range of image-text tasks, including discriminative tasks and generative tasks such as visual captioning and question-answering. (3) Our model comes with AS-1B the largest dataset with open-world panoptic semantics. Data and models feed each other in the data engine, iteratively improving the model performance, data scale and diversity.

In summary, our contributions are three folds:

(1) We propose a new large-scale dataset (AS-1B) for open-world panoptic visual recognition and understanding, using an economical semi-automatic data engine that combines the power of off-the-shelf vision/language models and human feedback. As reported in Table 1, we have 159 times more semantic tags and 33 times more regions compared with its counterparts.

(2) Based on the dataset, we develop a unified vision-language foundation model (ASM) for open-world panoptic visual recognition and understanding. Aligning with LLMs, our ASM supports versatile image-text retrieval and generation tasks, demonstrating impressive zero-shot capability.

(3) We evaluate our model on a representative vision and vision-language tasks. Our ASM outperforms CLIP (Radford et al., 2021) by 10.7 and 13.4 (mAP) on COCO (Lin et al., 2014) and LVIS (Gupta et al., 2019) in zero-shot region recognition tasks. When trained with AS-1B (region-level data) and LaionCOCO (Schuhman et al., 2022) (image-level data), our model achieves superior zero-shot and fine-tuned performance compared to recent image-level (Dai et al., 2023; Wang et al., 2023b; Huang et al., 2023) and region-level (Yu et al., 2017; Wu et al., 2022; Peng et al., 2023) VLLMs.

## 2 RELATED WORK

**Datasets for Visual Recognition and Understanding.** As one of the three pillars of deep learning, datasets play a critical role in the advancement of deep learning models, especially in the field of visual recognition and comprehension. Prior to the era of large-scale models, datasets (Deng et al., 2009; Lin et al., 2014; Goyal et al., 2017) are primarily closed-world or have limited data scale. Additionally, datasets like Visual Genome (Krishna et al., 2017) and Visual7W (Zhu et al., 2016) integrate visual location and understanding, offering more comprehensive tasks to describe the visual

Table 1: **Comparison with popular vision and vision-language datasets**. "#" denotes the number of something. "Open" and "Closed" means Open-World and Closed-Set respectively. We see that the proposed AS-1B dataset has a significantly larger data scale and diversity than its counterparts.

| Type | Dataset | #Images | #Regions | #Concepts | #Tokens | Location | Semantic |
|---|---|---|---|---|---|---|---|
| Image-Level | ImageNet-22K (Deng et al., 2009) | 15M | – | 22,000 | – | – | Closed |
| | COCO Caption (Chen et al., 2015) | 0.1M | – | – | 8.4M | – | Closed |
| | CC12M (Changpinyo et al., 2021) | 12.4M | – | – | 250.9M | – | Open |
| | YFCC15M (Kalkowski et al., 2015) | 15M | – | – | 1.0B | – | Open |
| | COYO700M (Byeon et al., 2022) | 700M | – | – | 15.0B | – | Open |
| | Laion-5B (Schuhmann et al., 2022) | 5B | – | – | 135.0B | – | Open |
| Region-Level | SA-1B (Kirillov et al., 2023) | 11M | 1.1B | – | – | Open | – |
| | COCO (Lin et al., 2014) | 0.1M | 0.9M | 80 | – | Closed | Closed |
| | LVIS (Gupta et al., 2019) | 0.1M | 1.5M | 1,203 | – | Closed | Closed |
| | Open Images (Kuznetsova et al., 2020) | 1.5M | 14.8M | 600 | – | Closed | Closed |
| | BigDetection (Cai et al., 2022) | 3.5M | 36.0M | 600 | – | Closed | Closed |
| | V3Det (Wang et al., 2023a) | 0.2M | 1.5M | 13,029 | – | Closed | Closed |
| | Visual Genome (Krishna et al., 2017) | 0.1M | 0.3M | 18,136 | 51.2M | Open | Open |
| | AS-1B (ours) | **11M** | **1.2B** | **3.5M** | **132.2B** | Open | Open |

world. However, these datasets have limited semantics and fail to encompass diverse scenarios in the open world, which hinders the generalization ability of models. To achieve open-world capability, CLIP (Radford et al., 2021) and ALIGN (Jia et al., 2021) propose training models using web-scale image-text pairs collected from the internet. Subsequent works, such as Laion (Schuhmann et al., 2021; 2022), COYO-700M (Byeon et al., 2022) and DataComp (Gadre et al., 2023), have also been introduced for open-source research. However, these approaches only include descriptions or question-answering pairs corresponding to the entire image, resulting in models struggling to accurately recognize and understand specific objects at the instance level. Recently, Kirillov et al. introduced SA-1B (Kirillov et al., 2023), which provides open-world location information such as boxes and masks but still lacks semantic details. So existing datasets cannot meet the requirements of data scale, open-world location, and semantics necessary for achieving visual AGI models, thus posing challenges in supporting human-like panoptic visual recognition and understanding.

**Models for Visual Recognition and Understanding.** Significant advancements have been made in the field of visual recognition and understanding in recent years. Previous methods (He et al., 2017; Chen et al., 2023b; 2022b; Kirillov et al., 2019) mainly concentrate on the close-set recognition while recent works begin to focus on the open world understanding. Subsequent works (Radford et al., 2021; Li et al., 2021; Yu et al., 2022) can recognize and understand the open world semantics while failing to capitalize on the powerful perception capabilities of existing powerful pre-trained models, increasing the cost of developing new models. In recent years, Large Language Models (LLMs) (Brown et al., 2020; OpenAI, 2023; Touvron et al., 2023) have demonstrated excellent performance across various tasks, showcasing their potential for semantic understanding, dialogue generation, programming, mathematical problem-solving, etc, which leads to the emergency of many LLM-based multimodal models (Li et al., 2023a; Zhu et al., 2023b; Liu et al., 2023a; Wang et al., 2023b; Liu et al., 2023b; Li et al., 2023b; Zhai et al., 2022; Tschannen et al., 2023). However, these works are only capable of recognizing the entire image, lacking the ability to comprehend specific regions within the image. Some concurrent methods (Chen et al., 2023a; Peng et al., 2023; Zhang et al., 2023) begin to focus on location-aware understanding. However, without the support of large-scale instance-level visual understanding data, the generalization ability of these models is still limited.

## 3 THE ALL-SEEING DATASET (AS-1B)

In this section, we introduce the All-Seeing-1B (AS-1B) dataset for open-world panoptic visual recognition and understanding. The dataset consists of 1.2 billion regions in 11 million images[2]. Each region is annotated with comprehensive information, including semantic tags, question-answer pairs, and captions. Compared with the previous visual recognition datasets like ImageNet (Deng et al., 2009) and COCO (Lin et al., 2014), visual understanding datasets like Visual Genome (Krishna et al., 2017) and Laion-5B (Schuhmann et al., 2022), *the proposed AS-1B dataset stands out due to its rich and diverse instance-level annotation and corresponding detailed object concepts and descriptions.*

---

[2]Images source from SA-1B (Kirillov et al., 2023)

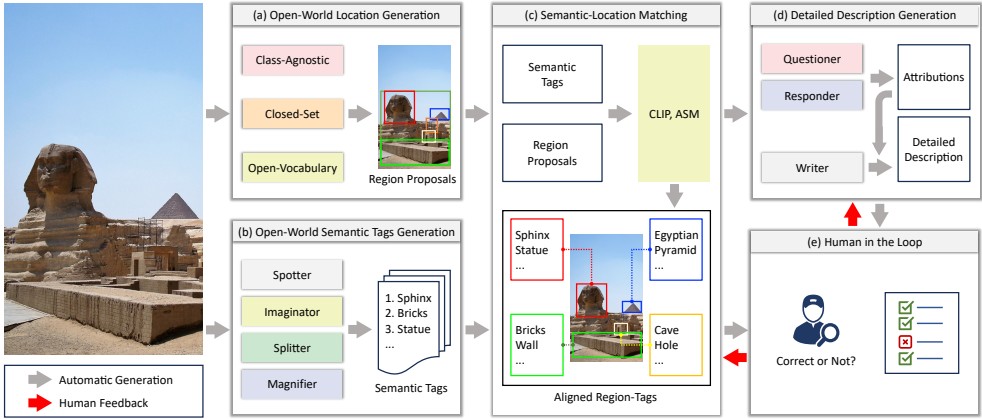

Figure 3: **Data engine for AS-1B dataset**. Our data engine consists of an automatic annotation pipeline (*i.e.*, (a), (b), (c), (d)) and a human verification stage (*i.e.*, (e)). We combine various powerful models to produce annotations for different regions, which are sampled and verified by human experts. Automated annotations are used together with human validation results to train region-aware models, which are then used in the automated annotation pipeline to improve data quality.

## 3.1 DATA ANNOTATION ENGINE

We develop a semi-automatic data engine that efficiently uses a wide range of state-of-the-art foundation models as annotators, reducing the enormous labeling cost to an acceptable level. As depicted in Fig. 3, the process of the data engine begins by generating noisy pseudo data using well-trained off-the-shelf foundation models (Kirillov et al., 2023; Wang et al., 2023c; Fang et al., 2023a; Li et al., 2022b; Radford et al., 2021; Lüddecke & Ecker, 2022; Chiang et al., 2023; Liu et al., 2023b) from diverse fields. Subsequently, a subset of these pseudo data are sampled to be verified and corrected by human annotators. After that, we pre-train our All-Seeing-Model (ASM) with the noisy pseudo data and finetune it with the human-verified data. Then we re-annotate the data with the aid of ASM. The process of annotation, verification, and fine-tuning are repeated to iteratively refine the annotation quality. By employing this "data-human-model" cycle, we can generate a large number of region-level annotations with exceptional quality. See Appendix E.3 for more details about the cycle.

As the core component of the data engine, the data generation pipeline consists of five steps as follows: (1) Creating open-world location (*e.g.*, bounding box, mask, point set) with an ensemble of state-of-the-art class-agnostic, visual grounding, and closed-set perception models (Kirillov et al., 2023; Li et al., 2022b; Wang et al., 2023c; Fang et al., 2023a); (2) Generating open-world semantic tags using the combination of image captioning models (Li et al., 2022a; Zhu et al., 2023b) and LLMs (Chiang et al., 2023); (3) Matching the semantic tags to proper regions with image-text aligning models (Radford et al., 2021; Lüddecke & Ecker, 2022); (4) Using LLM (Chiang et al., 2023) and VQA models (Liu et al., 2023b) to generate the attributions of each region based on the matched semantic tags; (5) Generating detailed captions based on the semantics and attributions of each region.

## 3.2 OPEN-WORLD LOCALIZATION

To obtain comprehensive locations of all instances in an image, we combine the results of state-of-the-art perception models from different fields, including (1) **class-agnostic model**: we adopt the SAM (Kirillov et al., 2023) to provide initial proposals of most objects in view. (2) **closed-set detection model**: we use InternImage-H (Wang et al., 2023c) and EVA-02 (Fang et al., 2023a) trained on BigDetection (Cai et al., 2022) and LVIS (Gupta et al., 2019), respectively, to detect the common-seen objects. (3) **grounding model**: we use GLIP (Li et al., 2022b) to ground open-world semantics generated by LLMs (see Sec. 3.3). All the bounding boxes are gathered together to ensure that all possible objects in view are covered. See Appendix B.1 for details of the gathering strategy.

## 3.3 OPEN-WORLD SEMANTIC

Manually labeling billions of regions for an open-world semantic description is impractical due to the enormous cost and time required. To remedy these challenges, we draw inspiration from the recent

advancements in Large Language Models (LLMs) and Visual Large Language Models (VLLMs). We leverage a series of LLMs (Chiang et al., 2023) and VLLMs (OpenAI, 2023; Liu et al., 2023a; Li et al., 2022a; Liu et al., 2023b; Wang et al., 2023b; Zhu et al., 2023b) as "semantic generators" and tap into their vast world knowledge and reasoning abilities for open-world semantic generation. These "semantic generators" can be specialized for producing short semantic tags or detailed descriptions, including attributes, question-answering pairs, and captions, based on specially-designed prompts.

**Semantic Tags**. To generate as many semantic tags as possible for a view, different instructions are employed to harness the diverse capabilities of LLMs and VLLMs, turning them into annotators with different focuses and skills. Specifically, we have (1) a **spotter**, which identifies major instances and provides an overview of the scenes, (2) a **imaginator** that leverages world knowledge to imagine plausible objects, (3) a **splitter** that divides complicated objects into parts, as well as (4) a **magnifier** which zooms on each region to produce region-specific candidates. These models complement each other to create a powerful system that can generate comprehensive open-world semantic tags for each region and the entire image. See Appendix B.2.1 for more details.

**Detailed Descriptions**. To provide detailed descriptions that include attributes and statuses of each region, we develop a pipeline that expands the region description using the open-world location and its matched semantic tags (see Sec. 3.4 for location-semantic matching). We utilize a series of skilled LLMs, including (1) a **questioner** that asks specific questions about the attributes or status of a given semantic tag; (2) a **responder** that provides the accurate answers for these questions based on the region's content; and (3) a **writer** responsible for composing a detailed caption for each region, according to the generated semantic tags, attributes, and status. See Appendix B.2.2 for more details.

## 3.4 MATCHING LOCATION AND SEMANTIC

In the matching process, we employ a region-text aligning model to measure the similarity between a certain region and its semantic tag list. For each region, the semantic tag list is constructed by LLMs (*i.e.*, "spotter", "imaginator", "divider" and "magnifier") and closed-set object detectors. Initially, in the first iteration of the data engine, we use a CLIP model (Radford et al., 2021) for the region-text alignment, where the input is the cropped region. To make sure the semantic tag is matched with the major object in the bounding boxes, CLIPSeg (Lüddecke & Ecker, 2022) is also utilized to generate the mask for each candidate. The original CLIP confidence is then modulated with the corresponding mask area. In the subsequent iterations, we upgrade the CLIP model to our All-Seeing Model.

## 3.5 HUMAN VERIFICATION

Albeit efficient, annotations from the automated pipeline still contains some noise due to the cropping process, which might discard essential context information. Therefore, to enhance the data quality, we find it crucial to include human verification and implement a "data-human-model" loop to continuously improve the data quality. Specifically, the human annotators are asked to verify the annotations of semantic tags and question-answer pairs. For semantic tags, we simply the task for annotators by focusing on picking the incorrect ones from the top-5 candidates in each region. For question-answer pairs, a two-stage verification procedure is employed. In the first stage, human annotators are asked to annotate the pairs with one of four choices: correct answer, wrong answer, unanswerable question, or wrong semantic tag. Samples annotated as the latter two options are annotated with a rejection answer, while those annotated as "wrong answer" will be further corrected in the second stage. With such an annotation strategy, human annotators only need to verify the outputs of the model instead of writing any long sentence, which greatly reduces the annotation cost. See Appendix B.3 and B.4 for more details.

## 4 THE ALL-SEEING MODEL (ASM)

### 4.1 OVERAL ARCHITECTURE

*Our objective is to create a unified framework that supports contrastive and generative image-text tasks at both the image level and region levels.* By leveraging pre-trained LLMs and powerful vision foundation models (VFMs), this model demonstrates promising performance in discriminative tasks like region recognition, as well as generative tasks such as image captioning, region captioning, etc.

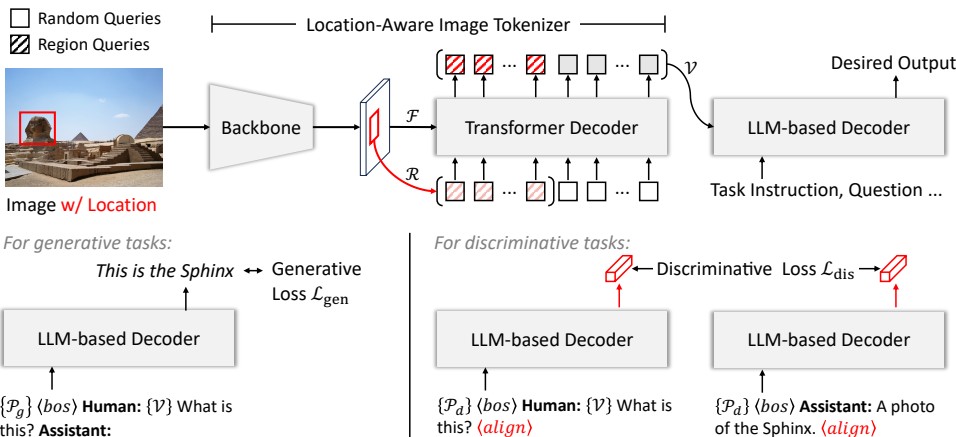

Figure 4: **Overview of the All-Seeing Model (ASM)**. ASM incorporates a location-aware image tokenizer to extract image-level and region-level features and a specific prompt design to handle both generative tasks and discriminative tasks using a unified architecture with shared parameters.

As illustrated in Fig. 4, our All-Seeing Model (ASM) comprises three key designs: (1) a **location-aware image tokenizer** extracting features from both the image and region levels based on the input image and bounding box, respectively. (2) a **trainable task prompt** that is incorporated at the beginning of the vision and text tokens to guide the model in distinguishing between discriminative and generative tasks. In the case of the discriminative task, a trainable align token is appended to the input sequence to gather the overall representation, and its embedding is then used in the matching process. (3) a **LLM-based decoder** that is utilized to extract vision and text features for discriminative tasks, as well as to auto-regressively generate response tokens in generative tasks.

The training objective of ASM is $\mathcal{L}_{\text{total}} = \mathcal{L}_{\text{gen}} + \mathcal{L}_{\text{dis}}$, where the generation loss $\mathcal{L}_{\text{gen}}$ is the same as the loss of GPT series (Radford et al., 2019) and the discriminative loss follows the contrastive loss of CLIP (Radford et al., 2021), where each region is treated as an image when calculating the loss.

## 4.2 LOCATION-AWARE IMAGE TOKENIZER

Here, we propose an extension of the Query Transformer (Li et al., 2023a) for a location-aware image tokenizer that conditions its queries on location information, such as bounding boxes. As depicted in Fig. 4, we first apply the ViT-g/14 (Fang et al., 2023b) to encode the input image and utilize RoIAlign (He et al., 2017) to extract the region features $\mathcal{R}$ from the image features $\mathcal{F}$ based on the given bounding box. Then, the flattened region features $\mathcal{R}$ are projected to the same shape as the randomly initialized query tokens $\mathcal{Q}'$. They are concatenated to form the location-aware query tokens $\mathcal{Q}$. Subsequently, these location-aware query tokens $\mathcal{Q}$ are passed through the Q-Former to extract output features from $\mathcal{F}$. Finally, the output features are projected to match the feature dimension of the LLM and are used as the soft prompt for subsequent decoding processes. Particularly, when no location information is provided, the bounding box is assumed to cover the entire image. This method guarantees a consistent approach for both local region and whole image tokenization.

## 4.3 LLM-BASED DECODER

To develop a unified LLM-based framework that can effectively handle both generation tasks and discriminative tasks, we utilize Husky-7B (Liu et al., 2023b) as our foundation language model.

**For generative tasks**, the input sequence comprises three types of tokens, including (1) learnable generative task prompts $\mathcal{P}_g \in \mathbb{R}^{M \times D_t}$, which informs the model that it should perform a generative task. (2) location-aware image tokens $\mathcal{V}$ that contain the extracted image-level and region-level information from the input image and (3) user prompt that expresses his/her requirements. Given such an input sequence, the LLM generates text tokens sequentially in an autoregressive manner until an end token $\langle eos \rangle$ is reached. An example prompt is shown in Appendix D.1.

Table 2: **Performance on the region-level captioning task.** "FS: 4" refers to the few-shot result with 4 samples. "-FT" denotes ASM with fine-tuning.

| Model | Zero-Shot | Visual Genome Meteor | Visual Genome CIDEr | RefCOCOg Meteor | RefCOCOg CIDEr |
|---|---|---|---|---|---|
| GRiT (Wu et al., 2022) | ✗ | 17.1 | 142.0 | 15.2 | 71.6 |
| SLR (Yu et al., 2017) | ✗ | - | - | 15.4 | 59.2 |
| SLR+Rerank (Yu et al., 2017) | ✗ | - | - | 15.9 | 66.2 |
| Kosmos-2 (FS: 4) (Peng et al., 2023) | ✗ | - | - | 14.1 | 62.3 |
| Kosmos-2 (Peng et al., 2023) | ✓ | - | - | 12.2 | 60.3 |
| ASM (ours) | ✓ | 13.0 | 46.8 | 15.0 | 48.8 |
| ASM-FT (ours) | ✗ | **18.3** | **148.7** | **21.8** | **107.8** |

**For discriminative tasks**, the input sequence of input image consists of soft prompt tokens that indicate task information, as well as vision tokens. Similarly, the input sequence of input text consists of the same soft prompt tokens as those of input image and text tokens that represent the corresponding region caption or object class name. In addition, we append a trainable align token $\langle align \rangle$ to each of the input sequences to extract the holistic representation of the current input sequence. During the process of region-text matching, we achieve image-text retrieval by simply computing the similarity of the embedding of the $\langle align \rangle$ token. The example prompts are provided in Appendix D.1.

## 5 EXPERIMENTS

We compare our ASM with CLIP-based and other VLLMs on representative vision tasks such as zero-shot region recognition, image caption, and region caption. In addition to standard metrics, we also use human evaluation for LLM-based models (see Appendix E.1). For the AS-1B dataset, a brief analysis has been provided in Fig. 2 and Table 1. The detailed analysis is present in Appendix C.

### 5.1 TEXT GENERATION

**Settings.** We evaluate the image-level caption ability of our model on Flickr30K (Young et al., 2014) and NoCaps (Agrawal et al., 2019) dataset. Following the common practice (Huang et al., 2023; Li et al., 2023a), we report the CIDEr (Vedantam et al., 2015) and SPICE (Anderson et al., 2016) metrics on these benchmarks. To assess the region-level caption ability, we also evaluate ASM on the Visual Genome (Krishna et al., 2017) and RefCOCOg (Mao et al., 2016). On the region caption task, we adopt both the Meteor (Banerjee & Lavie, 2005) and CIDEr metrics as our evaluation metrics, following Kosmos-2 (Peng et al., 2023). All of the metrics are computed by COCOEvalCap.

During training, a two-stage training process is employed. The first stage utilizes AS-1B (region-level) and LaionCOCO (image-level) for pre-training, while the second stage utilizes a subset of AS-1B that has been verified by human annotators, along with other high-quality data, for supervised fine-tuning. The fine-tuned ASM is denoted as ASM-FT. See Appendix D for more training details.

**Results.** For region-level captioning, as shown in Table 2, our ASM model surpasses the concurrent region-aware VLLMs, Kosmos-2 (Peng et al., 2023), by 2.8 points on the RefCOCOg dataset, under the zero-shot setting. After the second-stage fine-tuning, our ASM model has achieved a new record for referring expression generation on RefCOCOg. Besides, on the Visual Genome (VG) dataset, although the Meteor score of zero-shot ASM is inferior to GRiT (Wu et al., 2022), ASM-FT achieves significantly better results than GRiT given relevant data.

In addition, our model also excels at image-level captioning, as presented in Table 3, our ASM model demonstrates promising zero-shot performance on Flickr30K (Young et al., 2014) and No-Caps (Agrawal et al., 2019) dataset. Specifically, under the zero-shot setting, our model achieves a CIDEr score of 79.5 without the second-stage fine-tuning and 88.0 after the second-stage fine-tuning, which outperforms all the concurrent VLLMs, such as InstructBLIP (Dai et al., 2023), Shikra-13B (Chen et al., 2023a) and Kosmos-2 (Peng et al., 2023). Furthermore, on the NoCaps dataset, ASM also achieves comparable performance compared to the baselines. These results indicate that our ASM model retains a strong image-level comprehension ability while also being region-aware.

Table 3: **Zero-shot performance on the image-level captioning tasks.**

| Model | Zero-shot | Flickr30K | | NoCaps | |
|---|---|---|---|---|---|
| | | CIDEr | SPICE | CIDEr | SPICE |
| Flamingo-9B (Alayrac et al., 2022) | ✓ | 61.5 | - | - | - |
| SimVLM (Wang et al., 2022) | ✓ | - | - | 110.3 | 14.5 |
| BLIP (Li et al., 2022a) | ✓ | - | - | 113.2 | 14.7 |
| BLIP-2 (Li et al., 2023a) | ✓ | - | - | 121.6 | **15.8** |
| InstructBLIP (Dai et al., 2023) | ✓ | 82.8 | - | **123.1** | - |
| Shikra-13B (Chen et al., 2023a) | ✓ | 73.9 | - | - | - |
| Kosmos-1 (Huang et al., 2023) | ✓ | 67.1 | 14.5 | - | - |
| Kosmos-2 (Peng et al., 2023) | ✓ | 66.7 | - | - | - |
| ASM (ours) | ✓ | 79.5 | 17.6 | 107.7 | 14.6 |
| ASM-FT (ours) | ✓ | **88.0** | **18.8** | 116.9 | 15.6 |

Table 4: **Zero-Shot object recognition performance.** We report the zero-shot recognition performance on COCO and LVIS dataset. The ground-truth boxes are used for inference.

| Model | COCO | | | | LVIS | | | |
|---|---|---|---|---|---|---|---|---|
| | mAP | $AP_S$ | $AP_M$ | $AP_L$ | mAP | $AP_S$ | $AP_M$ | $AP_L$ |
| CLIP (Radford et al., 2021) | 58.9 | 50.7 | 70.4 | 58.3 | 47.1 | 40.3 | 59.2 | 57.4 |
| OpenCLIP (Ilharco et al., 2021) | 63.3 | 47.8 | 75.6 | 60.9 | 49.1 | 37.4 | 62.8 | 66.5 |
| R-CLIP (our baseline) | 68.6 | 61.4 | 75.4 | 79.3 | 54.8 | 49.3 | 60.6 | 66.6 |
| ASM (ours) | **69.6** | **63.7** | **77.3** | 72.2 | **60.5** | **55.8** | **67.3** | **69.3** |

## 5.2 ZERO-SHOT REGION RECOGNITION

**Settings.** We use zero-shot region recognition to evaluate the region-text alignment ability of our model. We use COCO (Lin et al., 2014) and LVIS (Gupta et al., 2019) detection dataset for evaluation. Since our current focus is not on object localization, we use the ground-truth boxes and use the model to predict the categories given the corresponding texts following RegionCLIP (Zhong et al., 2022). We report the mean Average Precision (mAP) metrics for this evaluation. The model is shared with text generation tasks, which is trained with a two-stage schedule.

**Results.** As shown in Table 4, both our baseline model R-CLIP (see Appendix D.3) and the proposed ASM achieve promising zero-shot region recognition performance. On the COCO dataset, R-CLIP outperforms the original CLIP by 9.7 mAP, and ASM further increases the performance by 10.7 mAP. On the more challenging LVIS dataset with 1,203 categories, R-CLIP outperforms CLIP by 7.7 mAP, and ASM achieves a more significant improvement of 13.4 mAP over CLIP. These results demonstrate the effectiveness of region-text data in AS-1B dataset and the proposed ASM in region-text alignment tasks. Notably, our ASM simultaneously performs caption and region recognition tasks with the same weight, showcasing its versatility and efficiency.

## 6 CONCLUSION

In this paper, we present the All-Seeing (AS) Project, which develops a comprehensive system for panoptic visual recognition and understanding in the open world, from both dataset and model perspectives. In terms of data, we elaborate a semi-automatic data engine consisting of an automatic annotation pipeline and a human verification step. Using this data engine, we annotated the AS-1B dataset comprising over 1 billion region-level comprehensive annotations, with controllable costs. From the model aspect, we propose a region-aware multi-modal large language model called the All-Seeing Model (ASM). The ASM utilizes a unified LLM decoder to model both region-text alignment and generation tasks. Leveraging the AS-1B dataset and other high-quality data, ASM achieves promising results on image and region-level tasks. We believe that the data engine, AS-1B dataset, and the ASM model proposed in the All-Seeing Project will inspire further research and development towards empowering artificial intelligence systems with an "all-seeing eye", enabling them to achieve a deeper understanding of the world.

ACKNOWLEDGMENTS

The work is supported by the National Key R&D Program of China (NO. 2022ZD0161300), and the National Natural Science Foundation of China (Grant No. 62376134).

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

APPENDIX OF "THE ALL-SEEING PROJECT: TOWARDS PANOPTIC VISUAL RECOGNITION AND UNDERSTANDING OF THE OPEN WORLD"

## A   DATASHEET FOR AS-1B DATASET

### A.1   MOTIVATION

Q1 **For what purpose was the dataset created?** Was there a specific task in mind? Was there a specific gap that needed to be filled? Please provide a description.

- AS-1B was created as the first large-scale dataset with comprehensive and detailed instance-level annotations. This dataset contains a wide range of open-world concepts, including over 3.5 million different semantic tags ranging from common categories (*e.g.*, human, backpack) to fine-grained or rare categories with attributes (*e.g.*, old metal latches), which serves as the data foundation to train a powerful model for open-world panoptic visual recognition and understanding. Before the curation of AS-1B, existing large-scale image-text pair datasets (Schuhmann et al., 2022; Byeon et al., 2022; Gadre et al., 2023) only include descriptions or question-answering pairs corresponding to the entire image. On the other hand, datasets like Visual Genome (Krishna et al., 2017) and Visual7W (Zhu et al., 2016) integrate visual location and understanding, offering more comprehensive tasks to describe the visual world. However, these datasets have limited data scale and fail to encompass diverse scenarios in the open world.

Q2 **Who created the dataset (e.g., which team, research group) and on behalf of which entity (e.g., company, institution, organization)?**

- This dataset is presented by OpenGVLab of Shanghai AI Laboratory.

Q3 **Who funded the creation of the dataset?** If there is an associated grant, please provide the name of the grantor and the grant name and number.

- This work was sponsored by Shanghai AI Laboratory.

Q4 **Any other comments?**

- No.

### A.2   COMPOSITION

Q5 **What do the instances that comprise the dataset represent (e.g., documents, photos, people, countries)?** *Are there multiple types of instances (e.g., movies, users, and ratings; people and interactions between them; nodes and edges)? Please provide a description.*

- Each instance in AS-1B represents a region in a certain image.

Q6 **How many instances are there in total (of each type, if appropriate)?**

- AS-1B consists of over 11 million images, 1 billion region-text pairs, 3.5 million open-world concepts, and 100 billion tokens of region-related question-answering and captions. A further overview of the statistics may be seen in the Table 1.

Q7 **Does the dataset contain all possible instances or is it a sample (not necessarily random) of instances from a larger set?** *If the dataset is a sample, then what is the larger set? Is the sample representative of the larger set (e.g., geographic coverage)? If so, please describe how this representativeness was validated/verified. If it is not representative of the larger set, please describe why not (e.g., to cover a more diverse range of instances, because instances were withheld or unavailable).*

- AS-1B is created based on the SA-1B dataset (Kirillov et al., 2023). From this collection of images, we extend the original annotations in SA-1B with a scalable semi-automatic data engine. Please see Section 3 for more details about the data engine.

Q8 **What data does each instance consist of?** *"Raw" data (e.g., unprocessed text or images) or features? In either case, please provide a description.*

- We provide 1.2 billion instance-level annotations. Each annotation consists of the following: an image file url; box coordinates in the format of $(x_1, y_1, x_2, y_2)$; semantic tag; three question-answer pairs; one detailed region caption.

Q9 **Is there a label or target associated with each instance?** *If so, please provide a description.*

- No, we do not define any label or target for the instances. Targets are task-dependent. AS-1B can be used for a variety of tasks such as region recognition (inputs = regions, targets = semantic tags), region captioning (inputs = regions, targets = captions), or region-level visual question answering (inputs = images and questions, targets = answers).

Q10 **Is any information missing from individual instances?** *If so, please provide a description, explaining why this information is missing (e.g., because it was unavailable). This does not include intentionally removed information, but might include, e.g., redacted text.*

- No.

Q11 **Are relationships between individual instances made explicit (e.g., users' movie ratings, social network links)?** *If so, please describe how these relationships are made explicit.*

- No.

Q12 **Are there recommended data splits (e.g., training, development/validation, testing)?** *If so, please provide a description of these splits, explaining the rationale behind them.*

- No.

Q13 **Are there any errors, sources of noise, or redundancies in the dataset?** *If so, please provide a description.*

- AS-1B is noisy by *design* since the annotations are produced by a semi-automatic data annotation pipeline. However, this pipeline operated in a "data-human-model" loop, iteratively refining data quality. Besides, we also provide a fully human-verified subset of AS-1B with over 800k instance-level annotations, which is much more clean.

Q14 **Is the dataset self-contained, or does it link to or otherwise rely on external resources (e.g., websites, tweets, other datasets)?** *If it links to or relies on external resources, a) are there guarantees that they will exist, and remain constant, over time; b) are there official archival versions of the complete dataset (i.e., including the external resources as they existed at the time the dataset was created); c) are there any restrictions (e.g., licenses, fees) associated with any of the external resources that might apply to a future user? Please provide descriptions of all external resources and any restrictions associated with them, as well as links or other access points, as appropriate.*

- This dataset is established based on SA-1B dataset (Kirillov et al., 2023). In response to sub-questions: (a) These image servers ensure stable access unless the SA-1B authors delete their images; (b) Yes, AS-1B archives all the annotations. For images, AS-1B only archives the URL and not the media content; (c) All restrictions follow the SA-1B dataset. We do not introduce any additional restrictions.

Q15 **Does the dataset contain data that might be considered confidential (e.g., data that is protected by legal privilege or by doctor–patient confidentiality, data that includes the content of individuals' non-public communications)?** *If so, please provide a description.*

- No.

Q16 **Does the dataset contain data that, if viewed directly, might be offensive, insulting, threatening, or might otherwise cause anxiety?** *If so, please describe why.*

- The annotations in this dataset are produced by a semi-automatic data engine. For the image component of AS-1B, we utilize high-resolution images from SA-1B. These images have undergone rigorous selection and privacy protection by Meta AI to ensure their suitability and compliance with privacy standards. For the text component of AS-1B, we have constrained the model not to generate offensive content and have removed annotations containing potentially offensive words or phrases[3].

---

[3]https://github.com/LDNOOBW/List-of-Dirty-Naughty-Obscene-and-Otherwise-Bad-Words

Q17 **Does the dataset relate to people?** *If not, you may skip the remaining questions in this section.*

- People might be found in the images or textual descriptions, but they are not the primary emphasis of the dataset.

Q18 **Does the dataset identify any subpopulations (e.g., by age, gender)?**

- We don't include any indicators of subpopulations as attributes for the region-text pairs, although it might be inferred in certain annotations.

Q19 **Is it possible to identify individuals (i.e., one or more natural persons), either directly or indirectly (i.e., in combination with other data) from the dataset?** *If so, please describe how.*

- No. The images in AS-1B are sourced from the SA-1B, where faces and vehicle license plates have been blurred in the released images.

Q20 **Does the dataset contain data that might be considered sensitive in any way (e.g., data that reveals racial or ethnic origins, sexual orientations, religious beliefs, political opinions or union memberships, or locations; financial or health data; biometric or genetic data; forms of government identification, such as social security numbers; criminal history)?** *If so, please provide a description.*

- No. The annotations in this dataset are generated by a semi-automatic data engine. Our model is not able to generate any sensitive content.

Q21 **Any other comments?**

- No.

### A.3 COLLECTION PROCESS

Q22 **How was the data associated with each instance acquired?** *Was the data directly observable (e.g., raw text, movie ratings), reported by subjects (e.g., survey responses), or indirectly inferred/derived from other data (e.g., part-of-speech tags, model-based guesses for age or language)? If data was reported by subjects or indirectly inferred/derived from other data, was the data validated/verified? If so, please describe how.*

- The annotations of AS-1B are generated by a semi-automatic data engine. As the core component of the data engine, the data generation pipeline consists of five steps as follows: (1) Creating open-world location (*e.g.*, bounding box, mask, point set) with an ensemble of state-of-the-art class-agnostic, visual grounding, and closed-set perception models (Kirillov et al., 2023; Li et al., 2022b; Wang et al., 2023c; Fang et al., 2023a); (2) Generating open-world semantic tags using the combination of image captioning models (Li et al., 2022a; Zhu et al., 2023b) and LLMs (Chiang et al., 2023); (3) Matching the semantic tags to proper regions with image-text aligning models (Radford et al., 2021; Lüddecke & Ecker, 2022); (4) Using LLM (Chiang et al., 2023) and VQA models (Liu et al., 2023b) to generate the attributions of each region based on the matched semantic tags; (5) Generating detailed captions based on the semantics and attributions of each region. See Section 3 for more details about the data engine.
- To verify the quality of the generated annotations, we asked 10 experts to annotate 1000 randomly sampled data from AS-1B with correct or wrong after each human-in-loop iteration separately. These annotations achieve an accuracy rate of 83.5%. In addition to the large-scale automatic annotations for over 1 billion regions, we will also release a clean version dataset, containing over 800k annotations that have been fully verified by human annotators. The accuracy of annotations from such a version is 95.7%.

Q23 **What mechanisms or procedures were used to collect the data (e.g., hardware apparatus or sensor, manual human curation, software program, software API)?** *How were these mechanisms or procedures validated?*

- We ran the data engine in Python, over 32 8-A100(80G) GPU machine. We validate our implementation by manually checking a few results of the data engine.

Q24 **If the dataset is a sample from a larger set, what was the sampling strategy (e.g., deterministic, probabilistic with specific sampling probabilities)?**

- AS-1B is created based on the SA-1B dataset. We annotate all images in the SA-1B.

**Q25 Who was involved in the data collection process (e.g., students, crowdworkers, contractors) and how were they compensated (e.g., how much were crowdworkers paid)?**

- Our data engine requires crowdworkers to verify a subset of the generated annotations. Specifically, human annotators are tasked with verifying the annotations of semantic tags and question-answer pairs. For semantic tag verification, we pay 0.02 dollars for each region. Regarding the verification of question-answer pairs, we pay 0.02 dollars for the first stage and 0.07 dollars for the second stage.

**Q26 Over what timeframe was the data collected? Does this timeframe match the creation timeframe of the data associated with the instances (e.g., recent crawl of old news articles)?** *If not, please describe the timeframe in which the data associated with the instances was created.*

- The licensed photos vary in their date taken over a wide range of years up to 2022

**Q27 Were any ethical review processes conducted (e.g., by an institutional review board)?** *If so, please provide a description of these review processes, including the outcomes, as well as a link or other access point to any supporting documentation.*

- We did not conduct a formal ethical review process via institutional review boards. However, as described in Q16, we employed several filtering mechanisms to try and remove instances that could be problematic.

**Q28 Does the dataset relate to people?** *If not, you may skip the remaining questions in this section.*

- People might be present in the images and descriptions, although they are not the sole emphasis of the dataset.

**Q29 Did you collect the data from the individuals in question directly, or obtain it via third parties or other sources (e.g., websites)?**

- We collect the data by annotating the images from the SA-1B dataset with our proposed semi-automatic data engine.

**Q30 Were the individuals in question notified about the data collection?** *If so, please describe (or show with screenshots or other information) how notice was provided, and provide a link or other access point to, or otherwise reproduce, the exact language of the notification itself.*

- For the image component of AS-1B, we utilize high-resolution images from SA-1B, which are licensed from a third party who provided appropriate representations regarding the collection of any notices and consents as required from individuals. For the text component of AS-1B, annotations are generated by a semi-automatic data engine, and the individuals in the image were not notified since their personal information has been hidden via facial blurring.

**Q31 Did the individuals in question consent to the collection and use of their data?** *If so, please describe (or show with screenshots or other information) how consent was requested and provided, and provide a link or other access point to, or otherwise reproduce, the exact language to which the individuals consented.*

- No. See Q30 for more details.

**Q32 If consent was obtained, were the consenting individuals provided with a mechanism to revoke their consent in the future or for certain uses?** *If so, please provide a description, as well as a link or other access point to the mechanism (if appropriate).*

- Users can contact the research team of the SA-1B dataset for image(s) removal. Besides, users can contact us to remove any annotation in our proposed AS-1B.

**Q33 Has an analysis of the potential impact of the dataset and its use on data subjects (e.g., a data protection impact analysis) been conducted?** *If so, please provide a description of this analysis, including the outcomes, as well as a link or other access point to any supporting documentation.*

- To eliminate any potential impact on people whose photos are included in the dataset, identifiable information (faces, license plates) has been blurred by the research team of the SA-1B dataset.

Q34 **Any other comments?**

- No.

## A.4 Preprocessing, Cleaning, and/or Labeling

Q35 **Was any preprocessing/cleaning/labeling of the data done (e.g., discretization or bucketing, tokenization, part-of-speech tagging, SIFT feature extraction, removal of instances, processing of missing values)?** *If so, please provide a description. If not, you may skip the remainder of the questions in this section.*

- The data engine operates in a "data-human-model" loop, iteratively refining data quality. Besides, we will also release a subset of AS-1B with over 800k annotations, which has been fully verified by human annotators.

Q36 **Was the "raw" data saved in addition to the preprocessed/cleaned/labeled data (e.g., to support unanticipated future uses)?** *If so, please provide a link or other access point to the "raw" data.*

- No.

Q37 **Is the software used to preprocess/clean/label the instances available?** *If so, please provide a link or other access point.*

- Yes, the data collection code will be open-sourced and accessible from the dataset website.

Q38 **Any other comments?**

- No.

## A.5 Uses

Q39 **Has the dataset been used for any tasks already?** *If so, please provide a description.*

- The AS-1B has been employed to train the All-Seeing-Model (ASM). As discussed in Section 5, ASM exhibits powerful performance in both region recognition and region captioning tasks, which strongly demonstrates the effectiveness of AS-1B.

Q40 **Is there a repository that links to any or all papers or systems that use the dataset?** *If so, please provide a link or other access point.*

- No.

Q41 **What (other) tasks could the dataset be used for?**

- The dataset could be used for a variety of region-level vision-and-language (V&L) tasks, such as region recognition, region captioning, and region-level visual question answering.

Q42 **Is there anything about the composition of the dataset or the way it was collected and preprocessed/cleaned/labeled that might impact future uses?** *For example, is there anything that a future user might need to know to avoid uses that could result in unfair treatment of individuals or groups (e.g., stereotyping, quality of service issues) or other undesirable harms (e.g., financial harms, legal risks) If so, please provide a description. Is there anything a future user could do to mitigate these undesirable harms?*

- No.

Q43 **Are there tasks for which the dataset should not be used?** *If so, please provide a description.*

- Our dataset should only be used for non-commercial academic research.

Q44 **Any other comments?**

- No.

### A.6 Distribution

Q45 **Will the dataset be distributed to third parties outside of the entity (e.g., company, institution, organization) on behalf of which the dataset was created?** *If so, please provide a description.*

- Yes, the dataset will be open-source.

Q46 **How will the dataset be distributed (e.g., tarball on website, API, GitHub)?** *Does the dataset have a digital object identifier (DOI)?*

- The data will be available through GitHub.

Q47 **When will the dataset be distributed?**

- 31/03/2024 and onward.

Q48 **Will the dataset be distributed under a copyright or other intellectual property (IP) license, and/or under applicable terms of use (ToU)?** *If so, please describe this license and/or ToU, and provide a link or other access point to, or otherwise reproduce, any relevant licensing terms or ToU, as well as any fees associated with these restrictions.*

- Apache 2.0 license

Q49 **Have any third parties imposed IP-based or other restrictions on the data associated with the instances?** *If so, please describe these restrictions, and provide a link or other access point to, or otherwise reproduce, any relevant licensing terms, as well as any fees associated with these restrictions.*

- AS-1B owns the metadata and release as Apache 2.0 license.
- We do not own the copyright of the images.

Q50 **Do any export controls or other regulatory restrictions apply to the dataset or to individual instances?** *If so, please describe these restrictions, and provide a link or other access point to, or otherwise reproduce, any supporting documentation.*

- No.

Q51 **Any other comments?**

- No.

### A.7 Maintenance

Q52 **Who will be supporting/hosting/maintaining the dataset?**

- Huggingface will support hosting of the metadata.
- OpenGVLab of Shanghai AI Laboratory will maintain the samples distributed.

Q53 **How can the owner/curator/manager of the dataset be contacted (e.g., email address)?**

- https://github.com/OpenGVLab/all-seeing

Q54 **Is there an erratum?** *If so, please provide a link or other access point.*

- No.

Q55 **Will the dataset be updated (e.g., to correct labeling errors, add new instances, delete instances)?** *If so, please describe how often, by whom, and how updates will be communicated to users (e.g., mailing list, GitHub)?*

- No. However, specific samples can be removed on request.

Q56 **If the dataset relates to people, are there applicable limits on the retention of the data associated with the instances (e.g., were individuals in question told that their data would be retained for a fixed period of time and then deleted)?** *If so, please describe these limits and explain how they will be enforced.*

- People may contact us to add specific samples to a blacklist.

Q57 **Will older versions of the dataset continue to be supported/hosted/maintained?** *If so, please describe how. If not, please describe how its obsolescence will be communicated to users.*

- We will only support and maintain the latest version at all times and a new version release of AS-1B will automatically deprecate its previous version.

Q58 **If others want to extend/augment/build on/contribute to the dataset, is there a mechanism for them to do so?** *If so, please provide a description. Will these contributions be validated/verified? If so, please describe how. If not, why not? Is there a process for communicating/distributing these contributions to other users? If so, please provide a description.*

- We welcome any contributions to AS-1B and we will announce updates regarding dataset extensions on GitHub. However, contributors must demonstrate the quality and harmlessness of the extended data annotations; otherwise, we will not accept these extensions.

Q59 **Any other comments?**

- No.

## B  DETAILS OF DATA ANNOTATION ENGINE

### B.1  OPEN-WORLD LOCALIZATION

Due to the incomparable score ranges of different models, directly using non-maximum suppression (NMS) to eliminate duplicated proposals from multiple resources is infeasible. Therefore, we develop an effective strategy that keeps all the semantics while removing highly overlapped regions. As shown in Alg. 1, the merging strategy works as follows: (1) We start by initializing the result region proposal set $\mathcal{R}$ with the class-agnostic bounding boxes generated by SAM. (2) When a set of region proposals $\mathcal{R}'$ from a new source (*e.g.*, closed-set/grounding detector) comes in, we calculate the Intersection over Union (IoU) between the regions in $\mathcal{R}'$ and $\mathcal{R}$. (3) If the IoU between a new region $r' \in \mathcal{R}'$ and an existing region $r \in \mathcal{R}$ is greater than a threshold $T_{\mathrm{IoU}}$, the region $r'$ is removed, and its closed-set/grounding tags are appended to the tag list of the matched region $r$. (3) Finally, the remaining low-IoU regions in $\mathcal{R}'$ along with their tags are added to $\mathcal{R}$. By employing this strategy, we sequentially combine the results of SAM, InternImage, EVA-02 and GLIP to obtain comprehensive location information for an image.

---

**Algorithm 1** Region Proposal Merging

**Input:**
    Existing region proposals $\mathcal{R}$
    New region proposals $\mathcal{R}'$
    IoU threshold $T_{\mathrm{IoU}}$

**Output:**
    Merged region proposals $\mathcal{R}$

1: **for** region $r' \in \mathcal{R}'$ **do**
2:     Calculate IoU between $r'$ and proposals in $\mathcal{R}$
3:     **if** maximum IoU $> T_{\mathrm{IoU}}$ **then**
4:         Merge semantic tags from $r'$ into the semantic tag of corresponding regions in $\mathcal{R}$
5:         Delete $r'$
6:     **else**
7:         Add $r'$ into $\mathcal{R}$
8:     **end if**
9: **end for**

---

### B.2  OPEN-WORLD SEMANTIC

Expanding on our brief description in Section 3.3, this section provides an illustration of the modules utilized to generate the semantic tags and detailed descriptions.

### B.2.1 Semantic tags

**Spotter**. This module aims to list the prominent and major objects present in the given image. To achieve this, we use MiniGPT4 (Zhu et al., 2023b) to provide an overall caption of the input image. From the generated captions, we extract noun phrases to serve as the semantic tags shared by all the regions in the input image. In addition, we also add an OCR detector (contributors, 2023) to detect the texts as semantic tags in the scenes. Note that the generated caption will also be passed to other annotators, which gives visual signal for the LLMs, serving as their eyes.

**Imaginator**. Although the "spotter" can find out the major objects in the scenes, it fails to identify many insignificant objects. To address this limitation, we develop an "imaginator" to further expand the semantic tag list with plausible imagination. The "imaginator" emulates human-like thinking. When provided with descriptions of a particular scene, humans can effortlessly imagine the potential objects present. For instance, if informed that an image depicts a group of children standing in a classroom, one may envision objects like "teacher", "blackboard", and "stationery". In our data engine, we utilize Vicuna (Chiang et al., 2023) to imagine possible objects in scenes based on the captions generated by the "spotter", and then extend the set using web search engines (Qiu et al., 2013). The "imaginator" excels at supplementing scene-specific object candidates, such as suggesting "airport stuff" instead of simply "person". This significantly enhances the concept diversity within this project.

**Splitter**. This model is proposed to divide the generated concepts into more fine-grained parts. We find that some region proposals only cover a part of the objects, such as the wing of a plane or the windshield of a car. However, most of the existing perception or caption models are not capable of detecting parts. To this end, we further instruct the Vicuna (Chiang et al., 2023) to divide the semantic tag into parts. For example, "building" will be decomposed into "roof", "door", "windows" and "walls". We tailor the prompt for LLM so that the model only divides the semantic tag that represents a concrete object into parts. LLM is instructed to ignore the semantic candidate that is non-physical or cannot be further divided, such as "water", "sky", etc.

**Magnifier**. Although hundreds of open-world semantic tags can be generated by the aforementioned annotators for each image, there still exists some regions whose semantics are absent from the generated tag lists. So we introduce a "magnifier" to zoom in on each region and add semantic tags for them. We simply crop the region and use a caption model to describe the cropped image, and then extract the noun phrases, which are used as the semantic candidates exclusive for the corresponding regions. In this model, we use BLIP (Li et al., 2022a) for efficiency.

### B.2.2 Detailed Descriptions

**Questioner**. Given semantic tag, to determine its commonly-used attributes, we use Vicuna (Chiang et al., 2023) as a questioner to generate three questions about the attributes or statuses. The prompt is shown below. In this way, we leverage the world knowledge and reasoning capabilities of LLMs to identify the most relevant attribute of an object.

---

**Instruction:** I will give you some objects. Please list 3 questions about the given objects. These questions must be answerable based on a photograph of the object and cannot rely on any outside knowledge. Some examples are listed as follows:

**Human**: Person
**Assistant**: Q1: What is the sex of this person? Q2: What is the hairstyle of this person? Q3: What is this person doing?

**Human**: {Semantic Tag}
**Assistant**:

---

**Responder**. After obtaining the questions related to a semantic tag, we employ Husky (Liu et al., 2023b), an LLM-based VQA model, to generate the responses to each question. The responses are generated in several sentences, taking into account the content of the region. An example prompt is shown below. This approach enables us to gather additional information about a region while preventing the inclusion of irrelevant content.

> **Instruction:** <ImageContent></img>
> **Human**: {Question for the image}
> **Assistant**:

**Writer**. Based on the question-answering pairs, we proceeded to use Vicuna (Chiang et al., 2023) to rephrase them into a single sentence, resulting in a detailed description of the region. The prompt used during annotation is shown below. It is notable that both the question-answering pairs from previous steps and the region captions from this step are valuable for visual recognition and understanding models.

> **Human:** Please paraphrase the following sentences into one sentence. {answer for question 1} {answer for question 2} {answer for question 3}
> **Assistant**:

### B.3 HUMAN ANNOTATION

Here, we introduce the details of human annotation process.

**Semantic tags.** We design a data sampling strategy and simplify the task for annotators by focusing on picking the incorrect ones from the top-5 candidates in each region. In the real world, concepts exhibit long-tail distribution as shown in Fig. 2. Therefore, many rare concepts will be missed if the region is randomly sampled for validation. To address this issue, we implement a concept-wise sampling strategy. Specifically, we collect a list of concepts in the first 1M images in the AS-1B dataset. From this list, we select most concepts for verification. We randomly sample 6 regions from the least frequent concepts and 90 regions from the concepts with the highest number of regions. During the human verification process, the semantic tag list for the sampled regions is provided to the annotators, who are then tasked with filtering out any incorrect tags.

**Visual Question-Answering Pairs.** Although using LLMs/VLLMs greatly reduces the annotation cost of generating visual question-answer pairs, there are still some issues that may introduce noise into the data. (1) The answer to the question is wrong since the VLLM is not perfect. (2) The generated question for the semantic tag may be unanswerable according to the given image content. (3) The semantic tag assigned to a region may be incorrect, leading to meaningless generated questions. For example, if a region containing a dog is wrongly labeled as a cat, asking about the color of the cat would be nonsensical.

To address these issues, we perform a two-stage verification procedure. In the first stage, human annotators are provided with the image, location (bounding box), and corresponding question-answer pairs. They are then asked to annotate the visual question-answer pair with one of four choices: correct answer, wrong answer, unanswerable question, or wrong semantic tag. Samples annotated as "correct answer" are retained, while those annotated as "wrong answer" are re-annotated with a correct answer generated by human annotators in the second stage. Samples annotated as "unanswerable question" or "wrong semantic tag" are annotated with a rejection answer, such as "This question is unanswerable according to the image" or "The object in this region is incorrectly labeled", respectively.

**Verification Review.** We engaged 50 human annotators to perform verification on the annotations generated by our model. To guarantee the quality of this verification process, we additionally request 10 experts to review the verified annotations. These experts are selected based on their domain knowledge and experience in annotation tasks. To streamline the process, we organize the regions requiring review into groups of 100. Each group is assigned to one expert, who checks the accuracy and consistency of the annotations within the group. Any package with an accuracy rate below 95% will be sent back for re-verification by another annotator. This review process double-checks the annotations, further ensuring their reliability and validity for our models.

### B.4 DATA ENGINE ITERATION

To continuously improve the data quality, we implement a "data-human-model" loop that maximizes the utilization of both human-verified data and models. As depicted in Alg. 2, the data engine iteration comprises three steps as follows: (1) The images are processed with the annotation pipeline which produces automatic annotations. (2) The ASM model is then trained using these coarse annotations, enabling it to perform both discriminative and generative tasks such as region-text matching and

---

**Algorithm 2** Data Engine

---

**Input:**

    Iteration Number $n$

    Images $\mathcal{I}$

    Models $\mathcal{M}$

    Annotation Pipeline $P(\mathcal{M}, \mathcal{I})$

**Output:**

    Annotations: $\mathcal{A}$

    Improved Models $\mathcal{M}$

1:  Generate initial annotation $\mathcal{A}_0$ by off-the-shelf models;

2:  Train ASM with $\mathcal{A}_0$, yield $\mathcal{M}_0$;

3:  $i \leftarrow 0$

4:  **while** $i < n$ **do**

5:     Perform Human verification on $\mathcal{A}_i$, yield $\mathcal{A}'_i$;

6:     Fine-tune $\mathcal{M}_i$ with $\mathcal{A}'_i$, obtain $\mathcal{M}_{i+1}$;

7:     Obtain Annotation $\mathcal{A}_{i+1}$ by $P(\mathcal{M}_{i+1}, \mathcal{I})$;

8:     $i \leftarrow i + 1$

9:  **end while**

---

region captioning. (3) The automatic annotations are sampled, reviewed, and corrected by human annotators, yielding high-quality human annotations. This verified data is then used to fine-tune the ASM model, thereby enhancing its performance. (4) The fine-tuned model is utilized to re-rank the semantic tags and generate more accurate region captions and answers. Repeat the third and fourth steps until the data quality meets the requirements. By following this data iteration process, we ensure continuous optimization of data quality, ultimately leading to superior results. In Appendix E.3, we demonstrate the effectiveness of this iteration process with quantitative experiments.

## C   DATA ANALYSIS

We conduct an in-depth analysis of our AS-1B dataset. We begin by showcasing the abundance of data in terms of quantity. Next, we explore the data diversity and open-world semantics captured in AS-1B. Finally, we thoroughly analyze the data quality of the initial automatic annotation pipeline and explain how we have improved it through data engineering and human feedback.

### C.1   DATA SCALE

**Statistics**. The AS-1B dataset consists of a vast collection of 1.2 billion region-text pairs extracted from 11 million images, encompassing 3.5 million distinct semantic tags. Regions in the dataset are categorized into five different resolution scales: tiny, small, medium, large, xlarge, and huge. As indicated in Table 5, the distribution of region resolutions follows a roughly normal distribution. Over half of the regions are on the medium or large scale. In Sec. 3.2, we utilize several region proposal generators, including SAM (Kirillov et al., 2023), InternImage (Wang et al., 2023c), EVA-02 (Fang et al., 2023a), and GLIP (Li et al., 2022b), to generate region proposals for the AS-1B dataset. Table 6 presents the proportion of regions provided by each model in the 1.2 billion regions. SAM generates 36.4% of the regions, while the other three models contribute to 63.6% of the regions. Therefore, although our dataset shares images with SA-1B (Kirillov et al., 2023) and has a similar number of regions, the actual regions are different due to the use of diverse region proposal generators.

Each region is also annotated with detailed question-answer pairs and a caption, which yields a total of 3.3 billion visual question-answering pairs and 1.2 billion detailed region captions. As seen in Table 7, the average token number of the answers is 16.91, while the average token number of the composited caption is 34.84. The total number of tokens in our detailed region captions amounts to approximately 42.2 billion. This extensive collection of detailed captions provides valuable textual descriptions of regions within the images.

Table 5: **Region statistics and semantic sources**. The percentage of semantic tags generated by different models at each resolution are reported. LLM/VLLMs (Chiang et al., 2023; Zhu et al., 2023b; Li et al., 2022a) contribute significantly to the semantic diversity of our dataset.

| Region Type | Area Range | Proportion | (V)LLMs | BLIP | InternImage | EVA-02 | GLIP |
|---|---|---|---|---|---|---|---|
| Tiny | $< 20^2$ | 4.2% | 33.8% | 16.5% | 24.6% | 25.1% | 0.0% |
| Small | $20^2 \sim 40^2$ | 8.7% | 34.5% | 14.3% | 24.6% | 25.9% | 0.7% |
| Medium | $40^2 \sim 100^2$ | 35.8% | 55.6% | 22.9% | 8.3% | 11.6% | 1.7% |
| Large | $100^2 \sim 200^2$ | 23.7% | 58.5% | 26.2% | 5.0% | 7.9% | 2.3% |
| Xlarge | $200^2 \sim 500^2$ | 18.3% | 62.6% | 27.1% | 3.0% | 4.3% | 3.0% |
| Huge | $> 500^2$ | 9.5% | 69.7% | 24.9% | 1.6% | 1.2% | 2.7% |
| All | − | 100% | 55.4% | 24.0% | 8.2% | 10.4% | 2.1% |

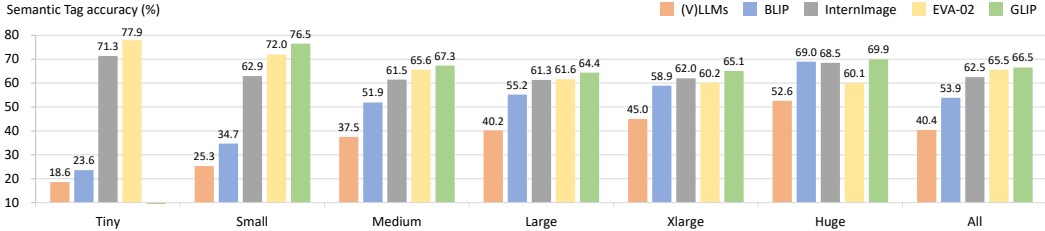

Figure 5: **The accuracy of semantic tags from different sources**. LLM/VLLMs (Chiang et al., 2023; Zhu et al., 2023b; Li et al., 2022a) show lower accuracy than other models, especially on low resolution regions.

**Comparisons**. When comparing the AS-1B dataset with popular datasets containing region-level annotations, AS-1B stands out with a significantly larger number of regions. It has about 33 times more regions than the current largest detection dataset, BigDetection (Cai et al., 2022). While AS-1B has fewer images compared to close-set classification datasets (Deng et al., 2009) or vision-language datasets (Schuhmann et al., 2022), it compensates with valuable region annotations. Additionally, AS-1B offers an abundant collection of detailed region annotations. Compared to the largest region-level dataset, Visual Genome (Krishna et al., 2017), AS-1B's detailed region annotation is about 1941 times larger than Visual Genome's 1.7 million pairs of VQA annotations and 222 times larger than its 5.4 million region captions.

## C.2 DATA DIVERSITY

**Statistics.** A distinctive feature of AS-1B is its vast inclusion of open-world concepts, demonstrated through two key aspects: 1) a large number of semantic tags and 2) long and informative detailed descriptions. Fig. 6 visually demonstrates the wide range of open-world concepts present in AS-1B. The dataset covers diverse categories, including fine-grained categories like "lynx", proper nouns such as "The Sphinxs", object parts like "charging cords", and attributes like "pink and white baby cribs". In Fig. 2, we display the frequency distribution of semantic tags, revealing a clear long-tail pattern. The most frequent semantic tags predominantly represent broad category names, while less frequent tags correspond to fine-grained category names or instances with specific attributes.

In Table 5, we analyze the sources of each semantic tag to understand how open-world concepts are enriched. We report the proportion of sources for the top-1 semantics in the semantic tags at different scales. The results reveal that 55% of the top-1 semantic candidates are from the LLM, while 24% originate from the BLIP (the "magnifier" in Sec. 3.3). Interestingly, only 19% of the top-1 candidates are generated from the closed-set detectors, InternImage, and EVA-02. This highlights that the majority of concepts in the AS-1B dataset are obtained from open-world sources, especially the LLMs and VLLMs.

However, although semantic tags generated by LLM/VLLMs can introduce comprehensive open-world semantic tag candidates, they may also introduce some semantic tags that do not exist in the image, thus giving rise to hallucinations. So we further analyze the accuracy of these retained semantic tags generated by LLM/VLLMs through human evaluation. Through human evaluation,

Table 6: **The proportion of region proposals generated by different models**. Only 40% regions are generated from SAM.

| Model | SAM | InternImage | EVA-02 | GLIP |
|---|---|---|---|---|
| Proportion | 36.4% | 20.5% | 22.5% | 20.6% |

Table 7: **The statistics of detailed description in AS-1B dataset**. The overall number of tokens reaches 132.2 billion.

| Type | Number | #Tokens | Average Tokens |
|---|---|---|---|
| Question | 3.3B | 34.6B | 10.50 |
| Answer | 3.3B | 55.4B | 16.91 |
| Caption | 1.2B | 42.2B | 34.84 |

Table 8: **The statistics of annotation accuracy**. AS-LoopX denotes the accuracy after the X-th human-in-loop iteration. AS-Human represents the accuracy of the subset of AS-1B which has been fully verified by human annotators.

| Stage | AS-Loop0 | AS-Loop1 | AS-Loop2 | AS-Loop3 | AS-Human |
|---|---|---|---|---|---|
| Acc for semantic tags | 54.8% | 70.2% | 76.7% | 80.6% | 95.3% |
| Acc for question-answer pairs | 54.8% | 75.0% | 80.3% | 83.5% | 95.7% |

the accuracy for the automatic data annotations is approximately 80.6%. For the data annotations that have undergone complete manual verification, the accuracy is 94.6%. Furthermore, the retained proportion of semantic tags ultimately produced from LLM/VLLMs is approximately 26.0%. These data demonstrate that, despite the fact that most semantic tags generated by LLM/VLLMs may not actually be present in the image, they do not introduce excessive noise after the semantic-location matching stage.

As for the detailed region caption, the VQA-based generation approach in AS-1B has proven advantageous, resulting in longer and more informative region descriptions. A more straight-forward way is to directly ask the VLLM to generate region captions. However, without guidance from semantic tags and questions, the model tends to output inaccurate information or hallucinations.

**Comparisons**. Instead of using fixed labels from a pre-defined set, the AS-1B dataset employs flexible and open-world semantic tags to label each region. Table 1 highlights that AS-1B contains a significantly larger number of semantic tags and concepts compared to close-set classification datasets or object detection datasets. For example, the number of semantic tags in AS-1B is approximately 159 times greater than the widely-used classification dataset ImageNet-22k (Deng et al., 2009), and it is 268 times larger than the category number in V3Det (Wang et al., 2023a).

## C.3 DATA QUALITY

**The Accuracy of Automatic Annotations**.     We asked 10 experts to annotate 1000 randomly sampled data from AS-1B with correct or wrong after each human-in-loop iteration separately and report the accuracy rates of our annotations for semantic tags and question-answer pairs after each human-in-loop iteration separately. As shown in Table 8, the introduction of human verification significantly improves the data quality, from 54.8% before human involvement to 75.0% after the first human-in-loop iterations. Moreover, as the number of loop iterations increases, the data annotation quality gradually improves. After the third loop, we are able to achieve an accuracy rate of 83.5%. In addition to the large-scale automatic annotations for over 1 billion regions, we will also release a clean version dataset, containing over 800k annotations that have been fully verified by human annotators. The accuracy of annotations from such a version is 95.3% for semantic tags and 95.7% for question-answer pairs.

As shown in Figure 5, we find that different models in the annotation pipeline exhibit complementary behavior. The LLM and BLIP models show lower accuracy for small regions as they are not robust for cropped low-resolution images. In contrast, close-set detectors perform better on these small regions, providing more accurate semantic candidates. For larger regions, LLMs and VLLMs become more accurate. Hence, the inclusion of close-set detectors can provide a trade-off between data quality and open-world semantics. This interplay of models contributes to the overall improvement of data quality in AS-1B.

**Consumption Analysis**. Here we focus on the consumption and efficiency of human verification in the context of the semi-automatic data engine we constructed. This approach significantly reduces the human labor required for data refinement compared with annotating all the data by humans. For verifying semantic tags, it takes approximately 10 seconds for one annotator to complete one region. Verifying every 1 million regions would take about 2,750 working hours. Considering a group of 50 annotators in our case, the entire verification process takes approximately 15 days. If we were to annotate all regions, the annotation consumption would become 1,000 times larger, approximately 42 years. Such a large-scale human annotation effort would be unaffordable.

Moreover, for detailed captions with longer texts, the verification process would take even longer, *e.g.*, 15 seconds for each VQA annotation. Therefore, for large-scale annotation involving billions of regions in our case, utilizing models to annotate data at scale and correcting the models' bias with limited human annotation proves to be both feasible and efficient.

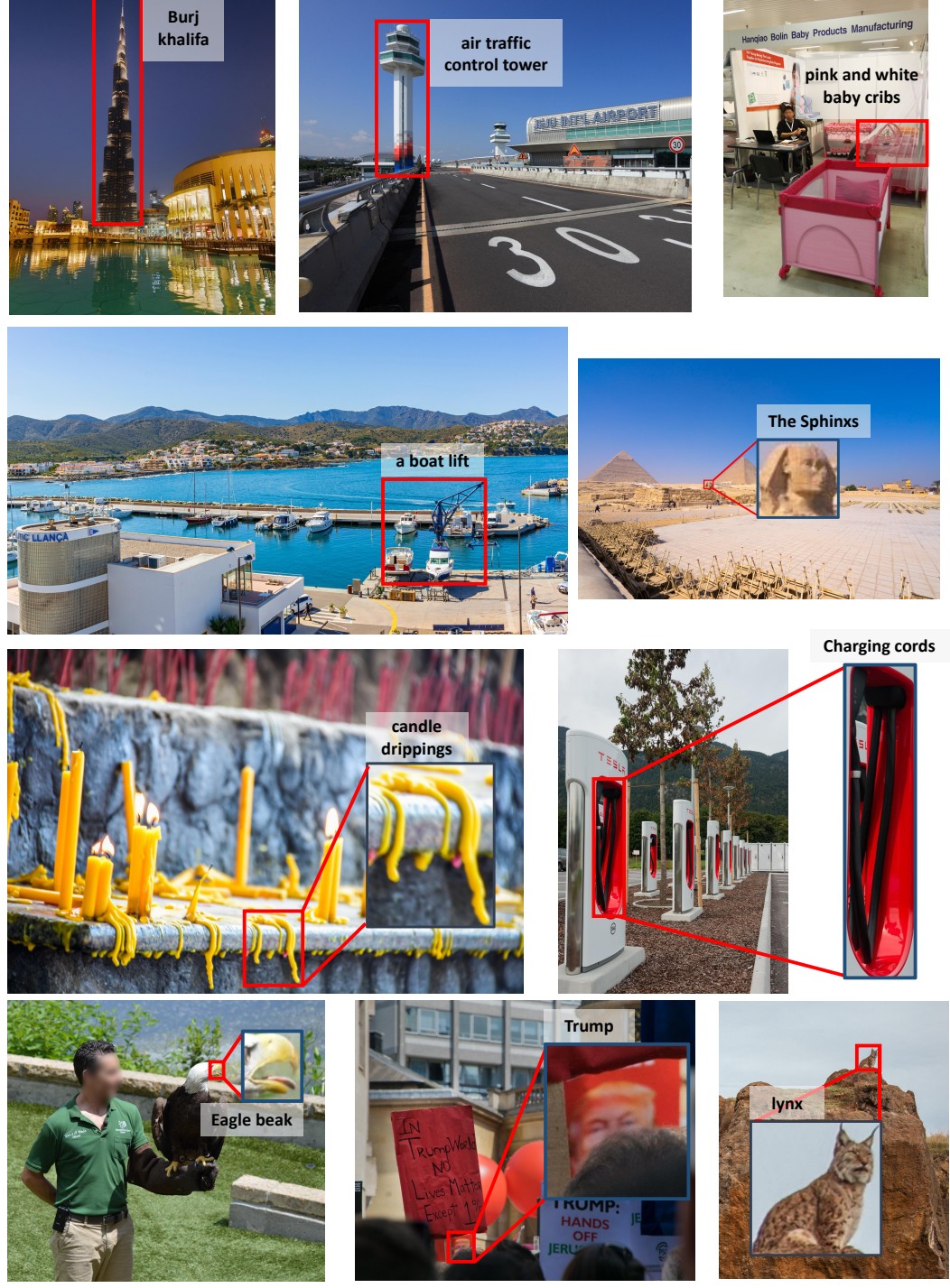

Figure 6: **Examples of the semantic tags**. Benefitting from the world knowledge of LLMs/VLLMs, the AS-1B dataset covers diversity semantic tags in the real world.

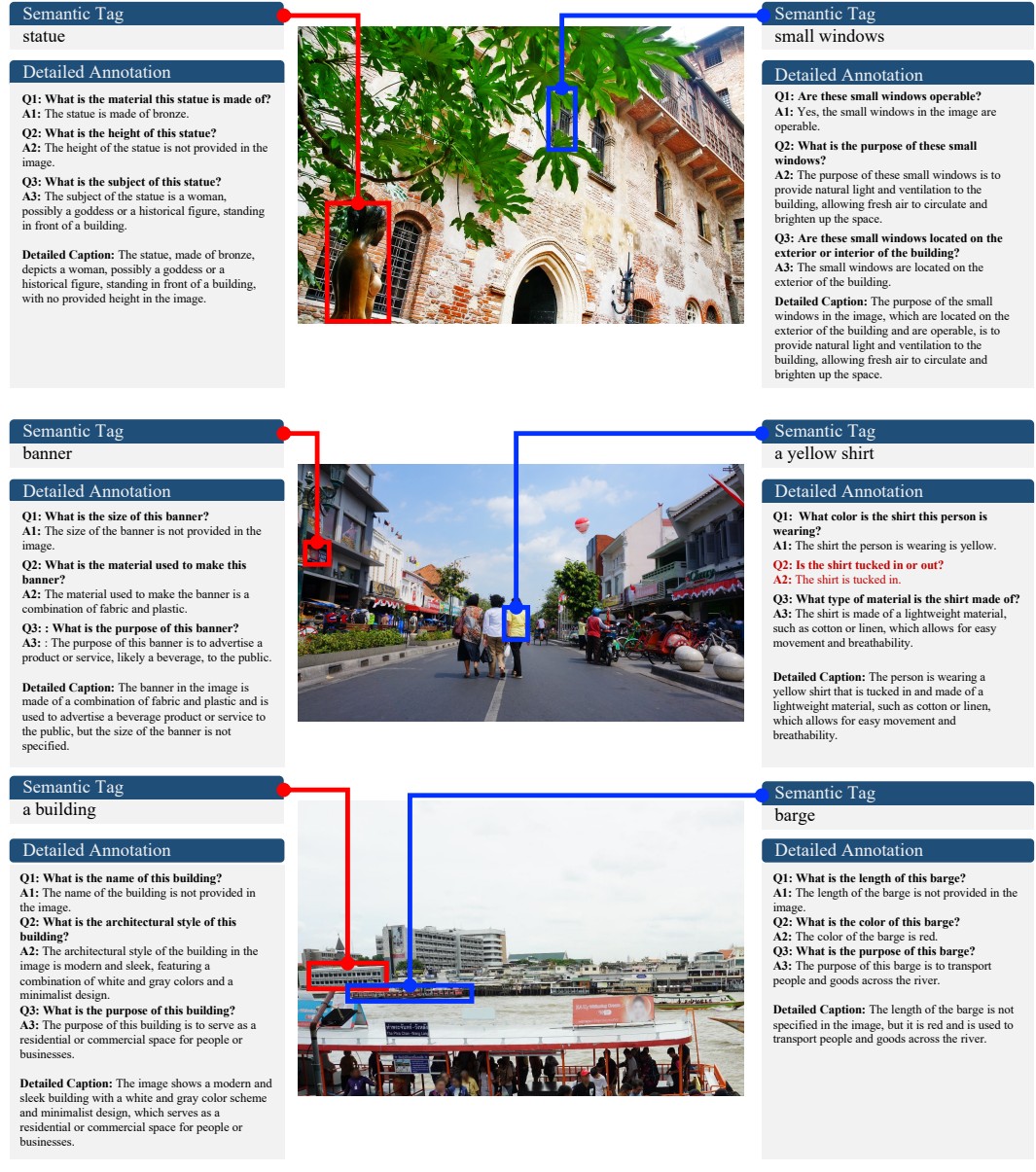

**Semantic Tag**
statue

**Detailed Annotation**

**Q1: What is the material this statue is made of?**
**A1:** The statue is made of bronze.
**Q2: What is the height of this statue?**
**A2:** The height of the statue is not provided in the image.
**Q3: What is the subject of this statue?**
**A3:** The subject of the statue is a woman, possibly a goddess or a historical figure, standing in front of a building.

**Detailed Caption:** The statue, made of bronze, depicts a woman, possibly a goddess or a historical figure, standing in front of a building, with no provided height in the image.

**Semantic Tag**
small windows

**Detailed Annotation**

**Q1: Are these small windows operable?**
**A1:** Yes, the small windows in the image are operable.
**Q2: What is the purpose of these small windows?**
**A2:** The purpose of these small windows is to provide natural light and ventilation to the building, allowing fresh air to circulate and brighten up the space.
**Q3: Are these small windows located on the exterior or interior of the building?**
**A3:** The small windows are located on the exterior of the building.

**Detailed Caption:** The purpose of the small windows in the image, which are located on the exterior of the building and are operable, is to provide natural light and ventilation to the building, allowing fresh air to circulate and brighten up the space.

**Semantic Tag**
banner

**Detailed Annotation**

**Q1: What is the size of this banner?**
**A1:** The size of the banner is not provided in the image.
**Q2: What is the material used to make this banner?**
**A2:** The material used to make the banner is a combination of fabric and plastic.
**Q3: : What is the purpose of this banner?**
**A3:** : The purpose of this banner is to advertise a product or service, likely a beverage, to the public.

**Detailed Caption:** The banner in the image is made of a combination of fabric and plastic and is used to advertise a beverage product or service to the public, but the size of the banner is not specified.

**Semantic Tag**
a yellow shirt

**Detailed Annotation**

**Q1: What color is the shirt this person is wearing?**
**A1:** The shirt the person is wearing is yellow.
**Q2: Is the shirt tucked in or out?**
**A2:** The shirt is tucked in.
**Q3: What type of material is the shirt made of?**
**A3:** The shirt is made of a lightweight material, such as cotton or linen, which allows for easy movement and breathability.

**Detailed Caption:** The person is wearing a yellow shirt that is tucked in and made of a lightweight material, such as cotton or linen, which allows for easy movement and breathability.

**Semantic Tag**
a building

**Detailed Annotation**

**Q1: What is the name of this building?**
**A1:** The name of the building is not provided in the image.
**Q2: What is the architectural style of this building?**
**A2:** The architectural style of the building in the image is modern and sleek, featuring a combination of white and gray colors and a minimalist design.
**Q3: What is the purpose of this building?**
**A3:** The purpose of this building is to serve as a residential or commercial space for people or businesses.

**Detailed Caption:** The image shows a modern and sleek building with a white and gray color scheme and minimalist design, which serves as a residential or commercial space for people or businesses.

**Semantic Tag**
barge

**Detailed Annotation**

**Q1: What is the length of this barge?**
**A1:** The length of the barge is not provided in the image.
**Q2: What is the color of this barge?**
**A2:** The color of the barge is red.
**Q3: What is the purpose of this barge?**
**A3:** The purpose of this barge is to transport people and goods across the river.

**Detailed Caption:** The length of the barge is not specified in the image, but it is red and is used to transport people and goods across the river.

Figure 7: **Examples of the detailed region annotations**. Visual question-answering pairs and captions are provided based on the semantic tags. Failure cases are marked in red.

# D INPLEMENTATION DETAILS

## D.1 MODEL DETAILS

The example prompts used by ASM are shown below. Prompt #1 is an example prompt for generative tasks, while Prompt #2 and #3 are example prompts for discriminative tasks.

> **Prompt #1:** "$\{\mathcal{P}_g\}$ $\langle bos \rangle$ **Human:** $\{\mathcal{V}\}$ What is this? **Assistant:**" ,

where the token number of task prompt $M$ is set to 5. $\langle bos \rangle$ represents the beginning of the sentence.

> **Prompt #2:** "$\{\mathcal{P}_d\}$ $\langle bos \rangle$ **Human:** $\{\mathcal{V}\}$ What is this? $\langle align \rangle$" ,

where $\mathcal{P}_d \in \mathbb{R}^{M \times D_t}$ represents the learnable task prompt used for discriminative tasks.

> **Prompt #3:** "$\{\mathcal{P}_d\}$ $\langle bos \rangle$ **Assistant:** A photo of the Sphinx. $\langle align \rangle$" .

It is notable that the learnable task prompt tokens and align tokens used in Prompt #2 and #3 are shared, while the task prompt tokens differ between generative tasks (Prompt #1) and discriminative tasks (Prompt #2 and #3).

## D.2 TRAINING DETAILS

The pre-training of the All-Seeing Model (ASM) involves three types of labels obtained from the AS-1B dataset, including region-level semantic tags, question-answer pairs, and detailed captions. The semantic tags are used for aligning regions with corresponding text, while the other annotations are used to train the text generation task. In addition, we also include LaionCOCO (Schuhman et al., 2022) in our pre-training process, since the image-level caption data from LaionCOCO is beneficial for ASM's ability to comprehend the whole images. For the supervised fine-tuning process, in addition to human-verified region annotations, we also utilize MiniGPT-4 (Zhu et al., 2023b), LLaVA-150k (Liu et al., 2023a), TextCaps (Sidorov et al., 2020), and COCO caption dataset (Chen et al., 2015) as image-level text generation data, along with VG (Krishna et al., 2017) and RefCOCOg (Mao et al., 2016) datasets as region-level text data. VQA datasets (Goyal et al., 2017; Hudson & Manning, 2019; Krishna et al., 2017; Chen et al., 2022a) are also utilized to enhance the vqa ability of ASM. In both of the stages, we optimize the Q-Former, soft prompt tokens and align tokens. In addition, LoRA (Hu et al., 2022) is employed to finetune the Vision Foundation Model (VFM) (Fang et al., 2023b) and Large Language Model (LLM) (Liu et al., 2023b).

We adopt a multi-task training approach that combines text generation and region-text alignment tasks to train our ASM. The batch size for text generation is set to 256, while for region text alignment it is set to 32,768. We employ the AdamW optimizer (Loshchilov & Hutter, 2019) with the $\beta_1$ of 0.9, the $\beta_2$ of 0.999, and the weight decay of 0. During training, the learning rate is initialized as $5 \times 10^{-4}$ and includes a linear warmup that lasts until the first 10% of training steps. The warmup is followed by a cosine decay strategy with a minimum learning rate of 0. The image resolution of ASM is set to $224 \times 224$ for pre-training and $364 \times 364$ for fine-tuning. We initialize the model parameters using Husky (Liu et al., 2023b) and train the model for one epoch. In addition, we also provide a second-stage fine-tuning setting to further improve the effectiveness of ASM. Specifically, we utilize high-quality multi-modal data MiniGPT-4 (Zhu et al., 2023b), LLaVA-150k (Liu et al., 2023a), and COCO caption dataset (Chen et al., 2015) as image-level text generation data, along with VG (Krishna et al., 2017) and RefCOCOg (Mao et al., 2016) datasets as region-level text data. Human-verified region annotations are also included. During fine-tuning, we set the learning rate to $5 \times 10^{-5}$ and apply a weight decay of 0. The other settings remain the same as during pre-training.

## D.3 BASELINE MODELS

To make comparison with recent popular multi-modality large language models (VLLMs) (Zhu et al., 2023b; Liu et al., 2023a; Li et al., 2023a) that only focus on processing the entire image, we crop a region from the image and input it to these models for region-level visual recognition

Table 9: **Human evaluation results on caption tasks.** We ask the users to select the caption that contains the most information regarding the image/region while does not producing any factual errors.

| Model | Visual Genome | | RefCOCOg | | Flickr30K | | NoCaps | |
|---|---|---|---|---|---|---|---|---|
| | Rate | Length | Rate | Length | Rate | Length | Rate | Length |
| Human | 47.8 | 13.6 | 10.3 | 6.3 | 30.0 | 16.0 | 27.3 | 15.1 |
| LLaVA (Liu et al., 2023a) | 4.3 | 110.8 | 15.4 | 100.6 | 17.5 | 114.0 | 9.1 | 108.4 |
| MiniGPT4 (Zhu et al., 2023b) | 8.7 | 110.9 | 15.4 | 113.5 | 14.2 | 114.6 | 13.6 | 101.0 |
| ASM (ours) | 39.2 | 34.5 | 46.1 | 110.8 | 38.3 | 121.4 | 50.0 | 115.9 |

and understanding. However, this cropping may result in the loss of some contextual information from the entire image. For better comparison, we implement a simple region-text contrastive model based on CLIP (Radford et al., 2021) as a baseline. The baseline model, named Region-Aware CLIP (R-CLIP), is equipped with a RoIAlign layer (He et al., 2017) on the feature maps obtained from the vision encoder in the CLIP model. To initialize the model weights, we leverage CLIP (Radford et al., 2021) (ViT-L/14) and then train the CLIP model on our AS-1B dataset. The model is trained for $10,000$ steps with a batch size of 32,768. Other training settings is the same as those of ASM. Unless otherwise specified, the image resolution of R-CLIP is set to $224 \times 224$.

# E    SUPPLEMENTARY EXPERIMENTS

## E.1    VISUAL CAPTIONING

As discussed in ChatCaptioner (Zhu et al., 2023a), using conventional image caption metrics such as Meteor (Banerjee & Lavie, 2005) and CIDEr (Vedantam et al., 2015) may not reliably evaluate relatively lengthy texts generated from LLM-based models. To better assess the text generation ability from a human perspective, we conducted a user study.

**Evaluation Setting.** In our user study, we involve a total of 5 participants to evaluate the performance of the All-Seeing Model (ASM) along with two other powerful VLLMs: MiniGPT4 (Zhu et al., 2023b), and LLaVA (Liu et al., 2023a). We evaluate image and region-level captioning. For the evaluation, we randomly select 20 samples from each of the Visual Genome, RefCOCOg, COCO, and Flickr30K datasets. Participants are asked to choose the most informative captions without any factual errors or hallucinations. Aside from model outputs, we also add the ground truth captions as options, which can be viewed as human outputs.

**Results.** The human evaluation results in Table 9 indicate that captions generated by our ASM are preferred over those from MiniGPT4 and LLaVA. While LLaVA and MiniGPT4 may produce longer captions for region-level tasks (VG and RefCOCOg), they often introduce over-association, hallucinations, and factual errors. In contrast, ASM generates captions with moderate length and more accurate information. On RefCOCOg, Flickr30K, and NoCaps datasets, ASM even outperforms human annotations with longer and more detailed captions. This is because human annotators tend to write short captions, while users prefer longer, detailed captions generated by ASM, which also contain fewer factual errors. For image-level generation tasks, ASM produces captions with similar lengths to those from MiniGPT4 and LLaVA but is more frequently favored by users.

The results clearly demonstrate the effectiveness of ASM and the AS-2B dataset. The VQA-based annotation pipeline provides region-specific information with less irrelevant content, reducing the occurrence of hallucinations. Moreover, human verification further enhances the data quality, leading to significantly better performance on region-level tasks.

## E.2    VISUAL QUESTION ANSWERING

**Setting**. We evaluate our model on the general visual question answering (VQA) benchmarks, including VQAv2 (Goyal et al., 2017), OKVQA (Marino et al., 2019), and GQA (Hudson & Manning, 2019). Following BLIP2 (Li et al., 2023a), we report VQAScore (Goyal et al., 2017) metric on VQAv2 and OKVQA and Accuracy on GQA. Besides, considering the lack of widely recognized region-level VQA benchmarks, we perform the human evaluation to evaluate the region-level VQA ability of our ASM. We randomly select 100 question-anser pairs from the proposed AS-1B. Note

that these question-answer pairs are annotated by our proposed ASM and have not been manually verified. Then, 5 participants are involved, and each of them is asked to annotate 20 question-answer pairs with one of four choices: correct answer, wrong answer, unanswerable question, or wrong semantic tag (*e.g.*the question has nothing to do with the object in the region).

**Results**. For image-level visual question answering, as shown in table 10, our ASM model achieves comparable performance with recent Visual Large Language Models (VLLMs) (Dai et al., 2023; Peng et al., 2023). Specifically, on VQAv2 and OKVQA benchmarks, ASM achieves VQAScore of 73.1 and 48.6, respectively. On GQA benchmark, ASM achieves 59.0 accuracy. It is notable that the ground truth answers to the questions in these VQA

Table 10: **Quantitative results on the VQA task.**

| Model | VQAv2 | OKVQA | GQA |
|---|---|---|---|
| Flamingo-9B (Alayrac et al., 2022) | 51.8 | 44.7 | - |
| Flamingo-80B (Alayrac et al., 2022) | 56.3 | 50.6 | - |
| Kosmos-1 (Huang et al., 2023) | 51.0 | - | - |
| Kosmos-2 (Peng et al., 2023) | 51.1 | - | - |
| BLIP-2 (Li et al., 2023a) | 65.0 | 45.9 | 32.3 |
| InstructBLIP (Dai et al., 2023) | - | - | 49.5 |
| ASM-FT (ours) | 73.1 | 48.6 | 59.0 |

benchmarks are quite brief and consist of only several words, while our ASM model is proposed to annotate the region with detailed descriptions and tends to generate more detailed and complete sentences. So these traditional VQAScore are not proper to evaluate the ability of LLM-based VQA systems like ASM. For region-level visual question answering, as there are no standard benchmarks, we score our model via human verification. As presented in Table 8, our ASM achieves remarkable performance, which demonstrates the ability of ASM to generate detailed and accurate answers to region-level visual questions.

### E.3  DATA ENGINEERING

Here, we use quantitative results to show the impact of data quantity and data engineering, which is shown in Alg. 2. Considering the cost of the experiment, we use our baseline model R-CLIP. We use the Zero-shot object recognition metrics as in Sec. 5.2 to inspect the impact of data engineering, *i.e.*, we use the ground-truth boxes and use R-CLIP to determine the categories following RegionCLIP (Zhong et al., 2022).

**Data Scaling-Up**. We find that scaling up the semantic tags can be helpful for zero-shot region recognition. To verify this, we train our baseline R-CLIP with different amounts of semantic tags. As shown in Table 11, with more training data (from 1M to 5M images), the R-CLIP model attains better zero-shot object recognition performance.

**Data Cleaning.** Data cleaning and post-processing are important. In practice, the original data annotation pipeline outputs a total of 2.14 billion regions. We devise a simple data cleaning strategy: (1) we sample the top 100 regions with the highest CLIP score at different scales from each image in the AS-1B dataset and (2) we further re-rank the semantic candidates with CLIPSeg (Lüddecke & Ecker, 2022), as discussed in Sec. 3.4. This data cleaning process will compress the original 2.14B regions into 1.2B regions. As shown in Table 12, adding data cleaning can significantly improve the mAP by 6.0% and 7.5% on COCO and LVIS datasets.

**How human verification improves the model?** An important part of our data engine is to improve the model with human feedback. In this way, the improved model can be used to refine the initial data which is automatically generated. In this section, we investigate the effectiveness of the human verification process. We fine-tune the trained R-CLIP model with human-verified region annotations and find that a small number of human labels can significantly boost the model performance.

Specifically, to make the most of human labels, we utilized both the positive and negative candidates marked by the human annotators. When calculating the contrastive loss, for each region, we randomly selected one positive candidate and use all the unselected candidates as negative samples. Compared with the image-to-text part in the original CLIP-style contrastive loss, each region will be compared with more negative text samples. The unselected candidates can be viewed as valuable hard samples, indicating when the model will make mistakes.

As shown in Table 13, fine-tuning the model with human data can yield significant performance gain. This demonstrates that a small amount of human data can correct the model's bias and hard cases, thus improving performance. The effectiveness of human verification lays the foundation for data

Table 11: **Zero-shot object recognition performance (mAP)** with different training data scale.

| Data Scale | COCO | LVIS |
|---|---|---|
| 1M | 67.8 | 54.0 |
| 2M | 67.5 | **55.0** |
| 5M | **68.6** | 54.8 |

Table 12: **Zero-shot object recognition performance (mAP)** with and without data cleaning.

| Data Cleaning | COCO | LVIS |
|---|---|---|
| ✗ | 61.8 | 46.5 |
| ✓ | **67.8** | **54.0** |

Table 13: **Zero-shot object recognition performance (mAP)** with and without fine-tuning on human-verified annotations.

| Human Data | COCO | LVIS |
|---|---|---|
| ✗ | 67.8 | 54.8 |
| ✓ | **70.2** | **55.0** |

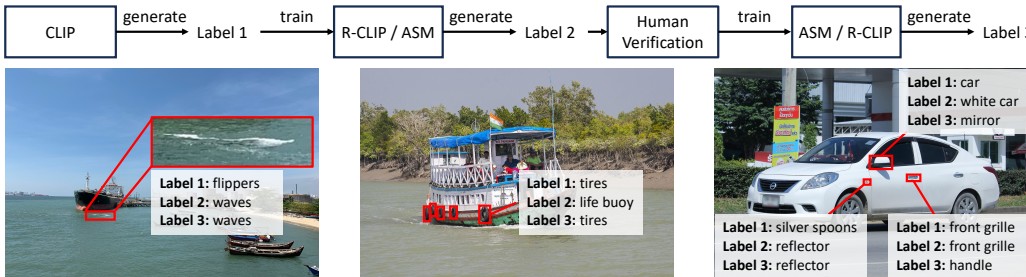

Figure 8: **Visualization of the data iteration process**. The iteration process improves the label accuracy. We visualize three types of models: (1) **Label 1**: labels produced the original CLIP; (2) **Label 2**: labels produced by R-CLIP or ASM, trained with **Label 1** as input data; (3) **Label 3**: labels produced by R-CLIP or ASM which is further tuned with human verification data.

quality improvement in the data engine iterations. To intuitively show the data quality improvements, we show the labeling results for CLIP as well as the outputs of R-CLIP before and after the human data fine-tuning in Fig. 8. The original CLIP is unreliable for small objects. Thanks to the data cleaning strategy, R-CLIP pre-trained on AS-1B data performs better in these small objects. However, it may fail to recognize some objects due to noisy labels, *e.g.*, labeling the tires hung by the boat as a "life buoy". The human data fine-tuning process can correct the pre-trained R-CLIP.

### E.4 ABLATION

Table 14: Ablations on captioning tasks.

| Model | VG | RefCOCOg | COCO Caption | Flickr30K | NoCaps |
|---|---|---|---|---|---|
| *zero-shot setting* | | | | | |
| Flamingo-9B (Alayrac et al., 2022) | - | - | 73.9 | 61.5 | - |
| Kosmos-2 (Peng et al., 2023) | - | 60.3 | - | 67.1 | - |
| Shikra (Chen et al., 2023a) | - | - | 117.5 | 73.9 | - |
| Emu (Sun et al., 2023) | - | - | 117.7 | - | - |
| ASM | 46.8 | 48.8 | 119.8 | 79.5 | 107.7 |
| ASM w/o LoRA | 42.6 | 45.3 | 123.0 | 81.7 | 111.0 |
| *SFT setting* | | | | | |
| GRiT (Wu et al., 2022) | 142.0 | 71.6 | - | - | - |
| BLIP (Li et al., 2022a) | - | - | 133.3 | - | 113.2 |
| BLIP2 (Li et al., 2023a) | - | - | 145.8 | - | 121.6 |
| InstructBLIP (Dai et al., 2023) | - | - | - | 82.8 | 123.1 |
| ASM-SFT | 148.7 | 107.8 | 140.2 | 88.0 | 116.9 |
| ASM-SFT w/o LoRA | 144.9 | 102.4 | 142.3 | 84.9 | 118.7 |

Considering the fine-tuning strategy utilizing LoRA (Hu et al., 2022) is different from other methods (Li et al., 2022a; 2023a; Dai et al., 2023), we supplement our results in Table 14, which includes outcomes without LoRA. Here, We report CIDEr. Note that the results under the SFT setting are fine-tuned with the COCO Caption dataset. We can observe that our model achieves powerful performance across the board in both zero-shot and finetuning settings. Besides, the results also reveal that our ASM without the LoRA fine-tuning strategy also demonstrates comparable performance with the counterpart tuned by LoRA. Since the ASM tune with LoRA achieves slightly better performance on region-level captioning tasks, we use this training strategy for human-in-loop iterations.

