# OpenReview forum: "The All-Seeing Project: Towards Panoptic Visual Recognition and Understanding of the Open World"
_ICLR.cc/2024/Conference — ICLR 2024 poster_

### Official Review · Reviewer_5eEK · 2023-11-02

**Soundness:** 3 good
**Presentation:** 3 good
**Contribution:** 3 good
**Rating:** 6
**Confidence:** 4

**Summary:**

This paper proposes a large dataset and model for detailed VQA and captions about image regions. The data engine involves the use of a combination of localization models, contrastive vision language models, and other LLMs/VLLMs, as well as humans in the loop to verify the outputs. The resulting dataset has 1.2B regions covering a wide range of 3.5M concepts. The authors also propose a model to ingest this data and handle both discriminative/generative tasks.

**Strengths:**

The paper addresses an important space of problems that existing vision and language foundational models are focused on image-level understanding, and there's a clear need to build region-level vision and language foundational model. The proposed dataset is based on the recent SA-1B dataset and extend it with semantic tags/QA pairs, and detailed caption, all of which can be useful to the community.

**Weaknesses:**

1. It'd be helpful to get some analysis on the quality of the final data after human verification. Appendix B.3 shows the accuracy of automatic annotation is around 50-60%. How much of the error is fixed by human verification, and how much is still there?

2. I'm wondering if it'd be better to set apart a high-quality split for region-level validation/testing of captioning. Existing dataset don't seem to serve this purpose very well e.g. RefCOCOg is not intended for region-level captioning. Visual genome is not a common benchmark for captioning evaluation either.

3. Image-level captioning in Table 3 is helpful, but not the focus of this work in my view. To make this more complete, it might be good to add COCO captions too.

4. To make a strong claim on region level understanding, I feel that the model should be able to predict regions from images rather than accepting regions as input. For example, in Table 4, it'd be more useful to have a simple ASM model that can predict regions without groundtruth box inputs.

5. It'd be great to have a region-level VQA benchmark as well since the dataset includes VQA. I see the image-level VQA results in Table 10, but that does not seem to capture the uniqueness of this data.

6. It'd be helpful to have some analysis on bias/fairness considerations.

**Questions:**

See weaknesses. My main concerns are with the data quality and evaluation benchmarks based on this dataset/model.

**Details Of Ethics Concerns:**

The dataset is built on SAM-1B using LLM and VLLMs. There could be potential bias/fairness concerns.

---

> ### Author Response · Authors · 2023-11-23
>
> Thank you for your time and expertise in the review process.
>
> **Q1: Analysis on quality of the final data after human verification.**
>
> **A1**: Thank you for the suggestions. We have conducted a thorough analysis of the data quality after human verification.
>
> We asked 10 experts to annotate 1000 randomly sampled data from AS-1B with correct or wrong after each human-in-loop iteration separately. In the table below, we report the accuracy rates of our automatic annotations after each human-in-loop iteration. As shown in the table, the introduction of human verification significantly improves the data quality, from 54.8% before human involvement to 75.0% after the first human-in-loop iterations. Moreover, as the number of loop iterations increases, the data annotation quality gradually improves. After the third loop, we are able to achieve an accuracy rate of **83.5%**.
>
> In addition to the large-scale automatic annotations for over 1 billion regions, we will also release a clean version dataset, containing over **800k** annotations that have been fully verified by human annotators. The accuracy of annotations from such a version is **95.7%**. We'll add more analysis of the annotations in the appendix of the next version of this paper.
>
> | Stage | AS-Loop0 | AS-Loop1 | AS-Loop2 | AS-Loop3 | AS-Human |
> | --- | --- | --- | --- | --- | --- |
> | Top-1 | 54.8 | 75.0 | 80.3 | 83.5 | 95.7 |
>
>
> **Q2: Is it better to set apart a high-quality split for region-level validation/testing of captioning? Existing datasets don't seem to serve this purpose very well.**
>
> **A2**: We argue that the existing datasets (e.g. Visual Genome and RefCOCOg) are good validation sets for region-level captioning. They have already been utilized to validate region captioning performance in many recent works, such as GRIT, Kosmos-2, GPT4RoI and GLaMM.
>
> On the other hand, the main contribution of this work is to create a large-scale dataset for visual recognition and understanding instead of evaluating the performance of existing models. The construction process of our dataset also refers to the construction process of datasets like SA-1B, Laion-5B, and CC-12M, which also do not have corresponding validation.
>
> We agree that the establishment of region-level evaluation benchmarks is quite worthy. However, we argue that it should be considered as independent research and it lies beyond the scope of our current study. As shown in recent benchmark works, such as MMBench, MME, MM-Vet and TouchStone, the cost of building a proper benchmark is large. We plan to incorporate region-level evaluation as our future work, providing a more detailed benchmark for ASM and comparison methods at that time. Thanks for your suggestions.
>
> **Q3: To make the experiment more complete, it might be good to add COCO captions too.**
>
> **A3**: Thank you for your advice. The tables below present the zero-shot and fine-tuned performance on COCO captions separately. We report CIDEr here. We can observe that our model achieves powerful performance on COCO captions in both zero-shot and finetuning settings. Since we focus on the zero-shot global captioning performance, we omit the performance of ASM on the coco caption dataset in the first version of our paper. Note that we finetuned the ASM with the COCO caption in the SFT stage.
>
> | Model | COCO Caption |
> | --- | --- |
> | Flamingo-9B | 73.9 |
> | Shikra | 117.5 |
> | Emu | 117.7 |
> | ASM | 119.8 |
>
> | Model | COCO Caption |
> | --- | --- |
> | BLIP | 133.3 |
> | BLIP2 | 145.8 |
> | ASM-SFT | 142.3 |
>
>
> **Q4: The model should be able to predict regions from images rather than accepting regions as input.**
>
> **A4**: The focus of this work is panoptic visual recognition and understanding. Perception like predicting regions from images is beyond our current scope. Please refer to the Common Questions Q1 for a detailed response to this question.
>
> **Q5: It'd be great to have a region-level VQA benchmark as well, since the dataset includes VQA.**
>
> **A5**: Thanks for your careful review. In Table 8 in the supplementary material and the new table shown in A1, we propose the human evaluation results for automatic annotations. Since these annotations, consisting of region-level VQA data, are generated by our ASM, these results can also be considered as the human evaluation results for region-level VQA.
> Besides, we do not consider the existing region-level VQA datasets, such as VGVQA, as benchmarks, because the annotated answers of existing VQA datasets are quite short and consist of only one word or phrase, lacking detailed information. However, as our project aims to generate a detailed description of the given region, these datasets are not proper to evaluate the VQA ability of our ASM.

---

> > ### Author Response · Authors · 2023-11-23
> >
> > **Q6: Analysis on bias/fairness considerations.**
> >
> > **A6**: Thank you for your advice. For the image component of AS-1B, we utilize high-resolution images from SA-1B. These images have undergone rigorous selection and privacy protection by Meta AI to ensure their suitability and compliance with privacy standards. For the text component of AS-1B, we will follow the approach of Laion-5B prior to open sourcing. All text content in the data will be scored and filtered using the open-source NSFW-Detector model. This process ensures that the content aligns with our standards and is appropriate for public use. In the final version of our paper, we will include an analysis of the scoring and filtering results. This will provide users with a comprehensive understanding of the content and the measures taken to ensure its quality and appropriateness.

---

### Official Review · Reviewer_WmLf · 2023-11-03

**Soundness:** 3 good
**Presentation:** 3 good
**Contribution:** 3 good
**Rating:** 6
**Confidence:** 3

**Summary:**

The papers presents a large-scale dataset collected using a semi-automatic data engine for open-world panoptic visual understanding. The dataset consists of 1 Billion + region annotations spanning semantic tags (3.5 Million +), question-answer pairs (3.3 billion) as well as detailed captions (1.2 billion). The paper proposes a VLLM called All-Seeing Model (ASM) trained on this dataset consisting of a location aware image tokeniser and a LLM based decoder. ASM achieves promising results on image and region-level captioning and recognition tasks.

**Strengths:**

- The paper does a great job explaining the details related to the dataset. The appendix contains several details that help understand the semi-automatic approach mentioned in the paper better (percentage of annotations from LLMs/VLLMs, accuracy of automatic annotations.
- The paper presents a fairly exhaustive benchmark (VQA, OK-VQA in supplementary). The paper also attempts to evaluate the model’s performance on region-based tasks like region-based visual question answering by conducting human studies.
- The paper also presents a good summary of many factors that are responsible for improving the performance of the model such as the role of data-engineering (D.3)

**Weaknesses:**

- Phrase Grounding Evaluation: The proposed method also missed an opportunity to leverage the dataset to learn the ability to ground language into the image by generating the bounding boxes corresponding to the text. I would have liked to see the models performance on the phrase grounding task on Flickr30K Entities.
- I think the paper misrepresents the state of the art in the community. For instance, the claim that current systems “are primarily focused on understanding images as a whole, rather than recognizing and comprehending individual instances within the scene” seems ungrounded, and several state of the art systems (e.g, Unified IO, including more recent ones like KOSMOS-2) show a fairly good understanding of the image on benchmarks that test this visual grounding like referring expressions, and phrase groundings. Since the authors compare and cite KOSMOS-2, for completeness the authors should also put the proposed dataset (AS-1B) in perspective of other comparable datasets such as GRiT which consists of region annotations for 90 million images.
- The paper uses LORA to fine-tune the LLM on various tasks COCO, VQA, etc which is different from other methods (BLIP, InstructBLIP, etc) that the method have compared against. This makes the evaluation unfair because these evaluations heavily penalise the peculiarities of the the evaluation protocol (one-word answers as opposed to natural language generation). Since methods like BLIP use a frozen LLM, it’s much harder for them to conform to the expected style of answers as opposed to the ASM which adapts the LLM using LORA.

**Questions:**

See weaknesses section

---

> ### Author Response · Authors · 2023-11-23
>
> Thank you for your efforts in the review.
>
> **Q1: Phrase Grounding Evaluation: The proposed method also missed an opportunity to leverage the dataset to learn the ability to ground language into the image by generating the bounding boxes corresponding to the text.**
>
> **A1**: Thanks for your careful review. The focus of this work is panoptic visual recognition and understanding. However, phrase grounding is more like a perception task, requiring models to predict the bounding box of the objects in the caption. Please refer to the Common Questions Q1 for the detailed response to this question.
>
> **Q2: The paper misrepresents the state of the art in the community.**
>
> **A2**: We did not misrepresent the state of the art in the community. We used the word “primarily” in that sentence to refer to works such as BLIP, LLaVA, etc., and in the Related Work section, we discussed these concurrent region-wise models, including ChatSpot, Kosmos-2, Shikra, and GPT4RoI. It is also important to note that without the support of large-scale instance-level visual understanding data, the generalization ability of these models is still limited, as discussed in the third paragraph of the introduction and the last sentence of the  related work.
>
> For Unified-IO, they focus on perception tasks instead of recognition tasks and do not conduct any experiment about region captioning or region VQA, so we don't have a discussion of them in the related work, and we'll make up a discussion of those in the next version of our paper.
>
> Besides, it is important to clarify that Kosmos-2 and our All-Seeing Project represent parallel efforts with different focus within this domain. Both GRIT and the AS-1B dataset feature region-level annotations and have demonstrated their efficacy in supporting the training of region-level MLLMs. However, there exist two significant distinctions between them: data format and supporting tasks.
>
> (1) The AS-1B dataset provides region-level annotations ranging from short semantic tags to entire conversations, while GRIT is limited to grounded image captions, and each region is only annotated with a short phrase.
>
> (2) their objectives diverge; AS-1B is designed to facilitate open-world recognition and understanding, encompassing diverse concepts and attributes using tools like Large Language Models (LLMs). In contrast, GRIT concentrates on perception and grounding tasks, where annotations primarily describe the context and predict the box coordinates for each instance.
>
> In essence, ASM and Kosmos-2 are complementary works. Our focus on integrating diverse concepts and attributes aligns with the open-world recognition goals, while GRIT's emphasis is on grounding tasks, and describing context with localization. Furthermore, apart from the automation annotations of over 1 billion regions, we will present a high-quality subset containing over 800k samples that are fully manually verified while GRiT only presents the automatic annotations. We will update these discussions in the revised version.
>
> **Q3: The fine-tuning strategy utilizing LoRA is different from other methods.**
>
> **A3**: Thanks for your careful review. We supplement our results in the table below, which includes outcomes without LoRA. We report the CIDEr here. The results reveal that without the LoRA fine-tuning strategy, our ASM also demonstrates comparable performance with the counterpart tune by LoRA. Since the ASM tune with LoRA achieves slightly better performance on region-level captioning tasks, we use this training strategy for human-in-loop iterations.
>
> | Model        | VG   | RefCOCOg | COCO Caption | Flickr30K | NoCaps |
> | ------------ | ---- | -------- | ------------ | --------- | ------ |
> | Flamingo-9B  | -    | -        | 73.9         | 61.5      | -      |
> | Kosmos-2     | -    | 60.3     | -            | 67.1      | -      |
> | Shikra       | -    | -        | 117.5        | 73.9      | -      |
> | Emu          | -    | -        | 117.7        | -         | -      |
> | ASM          | 46.8 | 48.8     | 119.8        | 79.5      | 107.7  |
> | ASM w/o LoRA | 42.6 | 45.3     | 123.0        | 81.7      | 111.0  |
>
>
> | Model            | VG    | RefCOCOg | COCO Caption | Flickr30K | NoCaps |
> | ---------------- | ----- | -------- | ------------ | --------- | ------ |
> | GRiT             | 142.0 | 71.6     | -            | -         | -      |
> | BLIP             | -     | -        | 133.3        | -         | 113.2  |
> | BLIP2            | -     | -        | 145.8        | -         | 121.6  |
> | InstructBLIP     | -     | -        | -            | 82.8      | 123.1  |
> | ASM-SFT          | 148.7 | 107.8    | 140.2        | 88.0      | 116.9  |
> | ASM-SFT w/o LoRA | 144.9 | 102.4    | 142.3        | 84.9      | 118.7  |

---

### Official Review · Reviewer_Wviv · 2023-11-25

**Soundness:** 3 good
**Presentation:** 3 good
**Contribution:** 4 excellent
**Rating:** 6
**Confidence:** 5

**Summary:**

In this paper, the authors present a large dataset and model for panoptic visual understanding, collectively named the "All-Seeing Project".
The dataset (AS-1B) contains more than one billion region-text pairs, where the text comprises semantic tags, question-answer pairs, and captions.
Text in AS-1B entails a rich vocabulary of visual concepts, e.g. the authors state the presence of 3.5 million unique semantic tags.
The authors design a scalable, semi-automatic data collection engine to collect AS-1B --
their pipeline is composed of several large vision models that generate region-text annotations,
and human annotator to verify the correctness of the generated annotations.
The authors train All-Seeing Model (ASM) using their dataset and show strong empirical performance on several downstream vision-language tasks.

**Strengths:**

1. AS-1B is a large dataset of region-text pairs, perhaps currently the largest of its kind.
2. The design choice of using the same images as SA-1B is excellent to mitigate ethical risks regarding copyright and privacy of users,
   as these images are meticulously verified by Meta AI, and released with a permissible license for research.
3. The proposed model (ASM) achieves strong empirical performance on several region-level visual understanding tasks.
4. The design of ASM allows it to be "composable" in a larger system that may include localization models like SAM.

**Weaknesses:**

**Note:**
I wrote my review a few days ago, but it comes late due to my oversight -- apologies for the delay.
In the interest of time, I have updated my initial asessment to incorporate the authors' rebuttal.
My outstanding concerns do not require the authors to run additional experiments.

I concur with the authors that the main contribution of this paper is a large-scale dataset.
AS-1B is currently the largest dataset of its kind (to the best of my knowledge)
and its availability will open new avenues for empirical study in the research community.
I view the model (ASM) as an important, however secondary contribution --
it serves as a baseline to provide a strong guarantee of data quality,
that training with AS-1B can yield strong empirical improvements across several task benchmarks.

My remaining concerns listed below are all geared towards ensuring a stronger guarantee of data quality
and repsonsible release of the dataset.

1. **Collection engine prone to hallucinations:**
The authors use large language models (LLMs) in the "imaginator" and "splitter" to produce semantic tags that are NOT conditioned on the visual content.
The imaginator produces plausible semantic tags that are _likely_ to occur, but not guaranteed to occur in images.
Is there is a way to quantify the amount of hallucination by checking the response of human annotators?
I suggest the authors to provide ample of qualitative examples in the paper showing the initial pool of semantic tags *before* they are assigned to the region proposals.

2. **Redundant caption annotations:**
The detailed caption of a region is produced by paraphrasing three question-answer pairs.
Based on the limited examples in the paper, the captions sound like a "dry" paraphrasing of the question-answer pairs (understandably so).
I wonder if having such redundancy contributes to the uniqueness of AS-1B, or simply adds redundancy and increases the size of dataset.

3. **Consider adding a datasheet:**
The authors should consider adding a datasheet () or a similar supplemental material outlining the characteristics of AS-1B.
For example, the Segment Anything paper includes a datasheet for SA-1B.
Datasheets serve as as a medium of communication between the authors (creators of the dataset) and future works (users of the dataset).
Many papers published in NeurIPS datasets track have datasheet templates which can be suitable in the ICLR format,
e.g. some image-text datasets like [LAION-5B](https://arxiv.org/abs/2210.08402) and [RedCaps](https://arxiv.org/abs/2111.11431).

4. **Needs a train/val split:**
I agree with the other reviewers' assessment that the authors should consider defining a train/val split for AS-1B.
If the authors do not define a split, different future works will regardlessly split AS-1B in ad-hoc ways and lead to inconsistencies.
I suggest the authors to "hold out" ~1% data as a validation set for the sake of consistency.

6. **Missing References:**
The proposed All-Seeing Model is trained with both, a generative and contrastive loss, to facilitate its use for generative (e.g. captioning) and discriminative (e.g. object recognition) tasks.
Due to the similarity in its architectural design, I believe the authors should cite a few prior works in their discussion to provide a broader context for the reader:

    - [CoCa: Contrastive Captioners are Image-Text Foundation Models](https://arxiv.org/abs/2205.01917) - trains with both objects as ASM.
    - [LiT: Zero-Shot Transfer with Locked-image text Tuning](https://arxiv.org/abs/2111.07991) - repurposes *any* image backbone to a contrastive image-text model.
    - [Image Captioners Are Scalable Vision Learners Too](https://arxiv.org/abs/2306.07915) - trains with generative objective first, then uses LiT.

**Questions:**

Please see weaknesses section.

**Details Of Ethics Concerns:**

The idea of using images from SA-1B is a major contributor to reducing the potential ethical risks arising from AS-1B. However, the paper lacks a discussion regarding the limitations of the proposed dataset and model. I suggest the authors to collect all the concerns that remain after this reviewer discussion and add a carefully worded section highlighting the limitations.
Broadly speaking, the AS-1B dataset
(1) is prone to the hallucinations produced by the constituent models,
(2) contains language that is uninformative, yet sounds "dry" (not a weakness, but better clarify upfront),
(3) may have annotation errors as a reasonable trade-off to reducing verification costs.

---

> ### Author Response · Authors · 2023-11-28
>
> Thank you for your valuable feedback and constructive comments.
>
> **Q1**: The authors use large language models (LLMs) in the "imaginator" and "splitter" to produce semantic tags that are NOT conditioned on the visual content. The imaginator produces plausible semantic tags that are likely to occur, but not guaranteed to occur in images.
>
> **A1**: Thank you for your careful review. The utilization of LLM/VLLMs to generate semantic tags is crucial because we aim for our candidate semantic tags to cover as many objects in the image as possible (see the first paragraph in Sec 3.3). According to our statistics, approximately **55.4%** of the retained semantic tags are derived from LLM/VLLMs, which emphasizes the importance of LLM/VLLMs in expanding open-world semantic tags.
>
> Although this step inevitably introduces some semantic tags that do not actually exist in the image, we will eliminate these hallucinations in the subsequent semantic-location matching stage and the human verification stage. During the semantic-location matching stage, we match semantic tags and filter out those that do not exist in the image. Specifically, in the semantic-location matching stage, we retain at most 5 semantic tags with high matching logit values for each region.
>
> Through human evaluation, the accuracy of these retained semantic tags generated by LLM/VLLMs is approximately **80.6%**.  In the 800k data annotations that have undergone complete manual verification, the accuracy is **94.6%**. Furthermore, the retained proportion of semantic tags ultimately produced from LLM/VLLMs is approximately **26.0%**. These data demonstrate that, despite the fact that most semantic tags generated by LLM/VLLMs may not actually be present in the image, they do not introduce excessive noise after the semantic-location matching stage.
>
> In summary, we argue that using LLM/VLLMs to generate semantic tags can effectively complement open-world semantics and enhance the coverage of semantic tags for objects in the image. Although this method inevitably introduces hallucinations, our subsequent semantic-location matching stage will strive to eliminate these hallucinations as much as possible. In the next version of the paper, we will include more analyses and qualitative examples for the LLM/VLLM-based semantic tag generation stage.
>
>
> **Q2**: Redundant caption annotations
>
> **A2**:  There might be a misunderstanding. The caption annotations are necessary and not redundant. Visual Question Answering (VQA) and image captioning are distinct yet critical tasks in the multi-modal field, each requiring specific types of data.
> For VQA, the annotated data consists of a question paired with a corresponding answer. In contrast, for image captioning, the annotations are descriptions of the images. The existing image descriptions are quite brief, which are not adequate for the demands of aligning multi-modal models in the era of Large Language Models (LLMs). Our dataset addresses this gap by including long detailed captions.
>
> However, there are challenges when creating long detailed captions : (1) the frequent inclusion of non-existent objects; (2) discrepancies between the caption's subject and the region's semantic tag. To remedy this, we crafted precise and detailed image descriptions based on brief captions and object attributes from VQA data, followed by manual correction. While the long detailed caption data does incorporate elements of the short caption and VQA data, they cannot be substituted with VQA data, nor vice versa. Despite leveraging LLMs can expedite the creation of these long detailed captions, manual verification and correction remain essential. There is no automated method to seamlessly interchange VQA and long detailed caption data. Hence, the long detailed caption data is irreplaceable and of significant importance for aligning multi-modal models in the LLM era, and is not redundant.
>
> **Q3**: Consider adding a datasheet
>
> **A3**: Thank you for your suggestion. We will add the datasheet to the supplementary materials of the next version of the paper.
>
> **Q4**: Needs a train/val split
>
> **A4**: Thank you for your suggestion. We will further manually select approximately 10k regions and their corresponding annotations from a total of 800k data annotations that have undergone thorough manual verification. Among these, 5,000 entries will be allocated for the validation set, and the remaining 5,000 entries will be designated for the test set.
>
> **Q5**: Missing References
>
> **A5**: We appreciate your recommendation. In the next version of our paper, we will incorporate the works you mentioned in both the related work and experiment sections.

---

### Author Response · Authors · 2023-11-23

Dear all reviewers:

We sincerely appreciate the reviewers for their time and effort in the review. First, we would like to highlight and clarify the key contributions of our research. Subsequently, we will address a common question, followed by detailed responses to each reviewer separately. We hope our responses could clarify existing doubts.



### Key Contributions Clarification

Here, we would like to emphasize that our core contribution is the AS-1B dataset, consisting of over 1.2 billion region annotations in various formats, such as semantic tags, question-answering pairs, and detailed captions. We emphasize that our proposed AS-1B is the first large-scale dataset with detailed instance-level annotations.

Compared to the existing large-scale image-text datasets, such as CC-12M and LAION-5B, our proposed AS-1B dataset contains instance-level annotations. On the other hand, the existing region-text datasets, such as COCO and RefCOCO, mainly concentrate on the perception task and annotate each region only with close-set label name or a short caption consisting of several words. Compared to these perception-oriented region-text datasets, our proposed AS-1B dataset annotates each region with open-world semantic tags, question-answering pairs, and detailed captions.

Furthermore, another crucial contribution of this work is the establishment of a scalable semi-automatic data engine, which significantly lowers the previously unaffordable expense of manually annotating a massive amount of open-world semantics. We ensure open-world localization with an ensemble of state-of-the-art class-agnostic, visual grounding, and closed-set perception models and generate open-world semantic tags using LLMs. Based on this data engine, we operate a “data-human-model” loop to annotate the AS-1B dataset at an affordable cost.

We emphasize that our proposed AS-1B dataset and the corresponding scalable data engine constitute our key contributions and the proposed ASM is only a validation of the effectiveness of our proposed dataset.



### Common Question

**Q1: The model should be able to predict regions from images rather than accepting regions as input or ground language into the image by generating the bounding boxes corresponding to the text.**

**A1**: The primary objective of this work is panoptic visual recognition and understanding. Predicting regions within images is beyond our current scope. This clarification can be divided into three key aspects:

(1) Perception and recognition are two related but distinct processes. The focus of this work is panoptic visual recognition and understanding, which requires the model to identify an object and describe it in detail given the specific region. This is distinct from perception, which focuses on predicting the locations (e.g bounding boxes) of objects.

(2) Effective user interaction in Visual Large Language Models (VLLMs) like GPT-4V depends on the system's ability to comprehend both pointer and language prompts. For instance, a user might ask the system to respond based on a specific region designated by a bounding box. The integration of pointer and language instructions enhances interaction accuracy and efficiency. This principle is foundational to the design of our proposed ASM, which is required to accept region and language prompts as input.

(3) Our work on panoptic recognition and understanding is orthogonal to perception models. If we need high-quality bounding boxes as pointer instructions, we can directly integrate a perception model (such as GLIP) with our ASM. In other words, locating objects is a readily attainable capability, and we don't need to repeat it in our ASM.

---

### Meta-Review · Area_Chair_KkJu · 2023-12-05

**Metareview:**

(a) Summary of Scientific Claims and Findings:
The paper introduces "All-Seeing Project," which includes a large dataset (AS-1B) and a model (ASM) for panoptic visual understanding. AS-1B, with over one billion region-text pairs and a rich vocabulary, is significant for its size and the diversity of its content, including semantic tags, question-answer pairs, and captions. The ASM model, trained using AS-1B, exhibits strong performance on various vision-language tasks. The dataset's collection process involves large vision models for annotation generation and human verification for accuracy.

(b) Strengths of the Paper:
1.AS-1B's size and comprehensiveness make it a potentially valuable resource for the research community.
2.ASM demonstrates strong empirical performance across multiple tasks.
3.The paper offers a thorough explanation of dataset creation and benchmarks.

(c) Weaknesses of the Paper:
1.Concerns about data quality: Potential issues with hallucinations in data collection and redundant captions need addressing.
2.The paper lacks a detailed discussion on limitations and ethical concerns.
3.Evaluation methods could be improved, such as including phrase grounding tasks and region-level VQA benchmarks.

**Justification For Why Not Higher Score:**

The paper is not awarded a higher score due to concerns over data quality, specifically regarding hallucinations in the dataset and potential redundancy in annotations. The lack of thorough discussion on limitations and ethical implications, as well as insufficient comparisons with relevant prior works, also contribute to the decision. Additionally, the evaluation methods could be more robust, including a more comprehensive analysis of the dataset's impact on various tasks.

**Justification For Why Not Lower Score:**

Despite its shortcomings, the paper presents a substantial contribution to the field with the AS-1B dataset and the ASM model. The scale and scope of AS-1B are noteworthy, and the ASM's strong performance across multiple benchmarks is commendable. The ethical approach to data collection and the detailed explanation of the dataset's creation and benchmarks are significant strengths that support the decision to not assign a lower score.

---

### Decision · Program_Chairs · 2024-01-16

Accept (poster)